Technical Report

# Integration of variant annotations using deep set networks boosts rare variant association testing

**Brian Clarke** [1,2,13] ✉, **Eva Holtkamp** [3,4,5,13], **Hakime Öztürk** [1],
**Marcel Mück** [2], **Magnus Wahlberg** [2], **Kayla Meyer** [2], **Felix Munzlinger**[2],
**Felix Brechtmann** [3,6], **Florian R. Hölzlwimmer**[3], **Jonas Lindner**[3], **Zhifen Chen**[7,8],
**Julien Gagneur** [3,5,6,9] ✉ **& Oliver Stegle** [1,10,11,12] ✉

Rare genetic variants can have strong effects on phenotypes, yet accounting for rare variants in genetic analyses is statistically challenging due to the limited number of allele carriers and the burden of multiple testing. While rich variant annotations promise to enable well-powered rare variant association tests, methods integrating variant annotations in a data-driven manner are lacking. Here we propose deep rare variant association testing (DeepRVAT), a model based on set neural networks that learns a trait-agnostic gene impairment score from rare variant annotations and phenotypes, enabling both gene discovery and trait prediction. On 34 quantitative and 63 binary traits, using whole-exome-sequencing data from UK Biobank, we find that DeepRVAT yields substantial gains in gene discoveries and improved detection of individuals at high genetic risk. Finally, we demonstrate how DeepRVAT enables calibrated and computationally efficient rare variant tests at biobank scale, aiding the discovery of genetic risk factors for human disease traits.

The recent arrival of population-scale whole-exome and whole-genome sequencing studies[1] vastly expands the potential to understand the genetic underpinnings of human traits. While genome-wide association studies (GWAS) on common variants have identified a compendium of trait-associated loci[2], mapping these largely noncoding variants to affected genes and addressing the typically subtle effect sizes remain challenging[3,4]. In contrast, rare variants can exhibit large effects[5], aiding the discovery of effector genes[6,7], the unraveling of molecular mechanisms underlying traits and, in turn, the identification of potent drug targets[8–10]. Of further medical relevance, modeling rare variant effects has recently shown promise for identifying individuals at high disease risk and in deriving polygenic risk scores (PRS) that generalize better across populations than those based only on common variants[11].

However, extending the GWAS strategy to rare variants must contend with a large number of low-frequency variants, leading to low statistical power due to sparsity and an increased multiple

[1]Division of Computational Genomics and Systems Genetics, German Cancer Research Center (DKFZ), Heidelberg, Germany. [2]AI Health Innovation Cluster, German Cancer Research Center (DKFZ), Heidelberg, Germany. [3]TUM School of Computation, Information and Technology, Technical University of Munich, Garching, Germany. [4]Helmholtz Association—Munich School for Data Science (MUDS), Munich, Germany. [5]Computational Health Center, Helmholtz Center Munich, Neuherberg, Germany. [6]Munich Center for Machine Learning, Munich, Germany. [7]Department of Cardiology, Deutsches Herzzentrum München, Technical University Munich, Munich, Germany. [8]Deutsches Zentrum für Herz- und Kreislaufforschung (DZHK), Partner Site Munich Heart Alliance, Munich, Germany. [9]Institute of Human Genetics, School of Medicine and Health, Technical University of Munich, Munich, Germany. [10]European Molecular Biology Laboratory, Genome Biology Unit, Heidelberg, Germany. [11]European Molecular Biology Laboratory, European Bioinformatics Institute (EMBL-EBI), Wellcome Genome Campus, Hinxton, UK. [12]Wellcome Sanger Institute, Wellcome Trust Genome Campus, Hinxton, UK. [13]These authors contributed equally: Brian Clarke, Eva Holtkamp. ✉e-mail: brian.clarke@dkfz-heidelberg.de; gagneur@in.tum.de; o.stegle@dkfz-heidelberg.de

testing burden. To compensate, rare variant association testing (RVAT) methods aggregate rare variants at the level of genomic regions, typically genes[12,13]. Such aggregation methods rely on information about which rare variants impact gene function, which typically cannot be inferred directly. Therefore, RVAT methods rely on functional annotations of variant effect[12,14–16], such as conservation scores or variant effect predictions for splicing, gene expression or protein structure[17–19], to prioritize putatively impactful variants.

Burden testing, a common RVAT strategy, relies on variant annotations to filter presumably uninformative variants and to weight informative ones. These weights are then aggregated into one or multiple alternative gene-level burden scores and tested for association with discrete or quantitative traits[20–25]. Complementary to burden tests, variance component tests, which can account for both protective and deleterious variants, also use annotations for filtering and weighting variants as part of a kernel function[13,26,27]. Recently proposed RVAT methods based on variance component tests convincingly demonstrated the added value of incorporating a broad spectrum of annotations either by conducting an omnibus test over different test types and annotations[28,29] or using specialized kernels tailored to different annotation types[30] (Supplementary Methods and Supplementary Table 1). However, like earlier methods, ad hoc variant filtering and weighting schemes remain integral components of existing workflows. The few RVAT methods that infer annotation weights from data and integrate multiple annotations are computationally prohibitively demanding and, in practice, limited in the number of annotations and in the flexibility of the scoring function that can be considered[31,32]. Finally, because of these limitations, none of these methods lends itself to phenotype prediction, thus limiting their utility for applications in personalized medicine (Supplementary Methods and Supplementary Table 1).

To address these issues, we present deep rare variant association testing (DeepRVAT). Our framework uses a deep set network for modeling traits through the integration of rare variant annotations. The model handles variable numbers of rare variants per individual, leverages dozens of continuous or discrete variant annotations and accounts for both additive and interaction effects. All model parameters are learned directly from the training data, minimizing the need for the ad hoc modeling choices that characterize existing methods. The trained DeepRVAT model gives rise to a single, trait-agnostic gene impairment scoring function. This offers several key advantages over variance component tests. First, it can be used for different genetic analyses, including rare variant association tests and the refinement of polygenic risk prediction to account for rare variant effects. Second, it can be easily integrated into single-marker genetic association testing tools, providing advantages such as maintaining calibration when testing for association with imbalanced binary traits. This is problematic for alternative methods using rich annotations. Finally, DeepRVAT provides large gains in computational efficiency over alternative methods. We provide pretrained DeepRVAT models for direct application on new datasets. If desired, the model can also be retrained efficiently from scratch, for example, to incorporate additional annotations.

We validate the model using simulations before applying it to 34 quantitative traits using whole-exome-sequencing data from 161,822 UK Biobank (UKBB) individuals, enhancing the number of discoveries compared to existing methods. Following this, we demonstrate superior replication of DeepRVAT associations in held-out individuals from UKBB. Moreover, we combine DeepRVAT gene impairment scores with polygenic risk models, which yields enhanced prediction accuracy for extreme-value phenotypes in UKBB traits. Finally, by integrating DeepRVAT with REGENIE[33], we demonstrate the utility of DeepRVAT in practical scenarios with related individuals, population structure and highly imbalanced binary traits. We apply DeepRVAT to 63 binary traits on a larger cohort of 469,382 UKBB individuals, yielding previously unknown associations with multiple diseases.

## Results

### A deep set network-based RVAT framework

DeepRVAT is an end-to-end genotype-to-phenotype model (Fig. 1a) that first accounts for nonlinear effects from rare variants on gene function (gene impairment module) to then model variation in one or multiple traits as linear functions of the estimated gene impairment scores (phenotype module). The gene impairment module (Fig. 1b) estimates a gene and trait-agnostic gene impairment scoring function that accounts for the combined effect of rare variants, thereby allowing the model to generalize to new traits and genes. Technically, a deep set neural network[34] architecture is used to aggregate the effects from multiple discrete and continuous annotations for an arbitrary number of rare variants. This architecture captures both linear additive and nonlinear effects and does not rely on a priori assumptions about the relevance of individual annotations, such as common assumptions about the relationship between allele frequency and effect size[20,22] (Methods).

To train DeepRVAT, an initial set of traits and corresponding associated genes (seed genes, specific to each trait) is required. The DeepRVAT software offers an integrated workflow that uses conventional rare variant association tests to identify seed genes.

DeepRVAT is trained end-to-end, optimizing the parameters of both the gene impairment module and trait-specific phenotype modules to predict trait variation from rare variants contained in the seed genes. To obtain robust gene impairment estimates while avoiding data leakage, we use cross-validation (CV) with multiple random initializations per fold (Extended Data Fig. 1; Methods).

Once estimated, the gene impairment scores can be used to test for genetic associations using established principles of conventional burden tests (Fig. 1c). The scores can also be used to predict phenotype from genotype, thereby providing a flexible way to derive a PRS[11,35] that accounts for rare variant effects (Fig. 1d). The training time of DeepRVAT scales linearly with the number of individuals, and association testing is highly efficient (Supplementary Fig. 1), thereby enabling applications to phenome-wide association studies (PheWASs) on large datasets such as UKBB.

To validate the model and assess the ability of DeepRVAT to learn properties of variant annotations from data, we initially considered a semi-synthetic dataset derived from the UKBB (Supplementary Fig. 2 and Supplementary Table 2). After confirming statistical calibration (Supplementary Fig. 3), we assessed the sensitivity of DeepRVAT and alternative RVAT methods to assumptions about the relevance of variant annotations. We find that DeepRVAT is robust to model misspecification, whereas alternative methods are sensitive to any mismatch between assumptions on the relevance of variant annotations and their simulated relevance (Supplementary Figs. 4 and 5).

### DeepRVAT improves RVAT yield and replication in UKBB

Next, we applied DeepRVAT to rare variants with minor allele frequency (MAF) <0.1% from whole-exome-sequencing (WES) data from the UKBB, considering 161,822 unrelated individuals of European ancestry (November 2020, 200k WES release[36]; Methods). We annotated 12,704,497 WES variants using MAF, variant effect predictor (VEP)[37] consequences, missense variant impact scores (SIFT[14], PolyPhen2 (ref. 16) and AlphaMissense[38]), omnibus statistical deleteriousness scores (CADD[15] and ConDel[39]), as well as predicted annotations for effects on protein structure (PrimateAI[18]), and aberrant splicing (SpliceAI[40] and AbSplice[17]). We further considered variant effect predictions for epigenetic markers in the encyclopedia of DNA elements (ENCODE)[41] and the Roadmap Epigenomics[42] projects (using low-dimensional projections of DeepSEA[19] predictions; Supplementary Fig. 6; Methods), as well as binding propensities for six RNA-binding proteins (selected predictions from DeepRiPe[43]; Methods). In total, this gave rise to 34 variant annotations (Supplementary Table 3 and Supplementary Fig. 7).

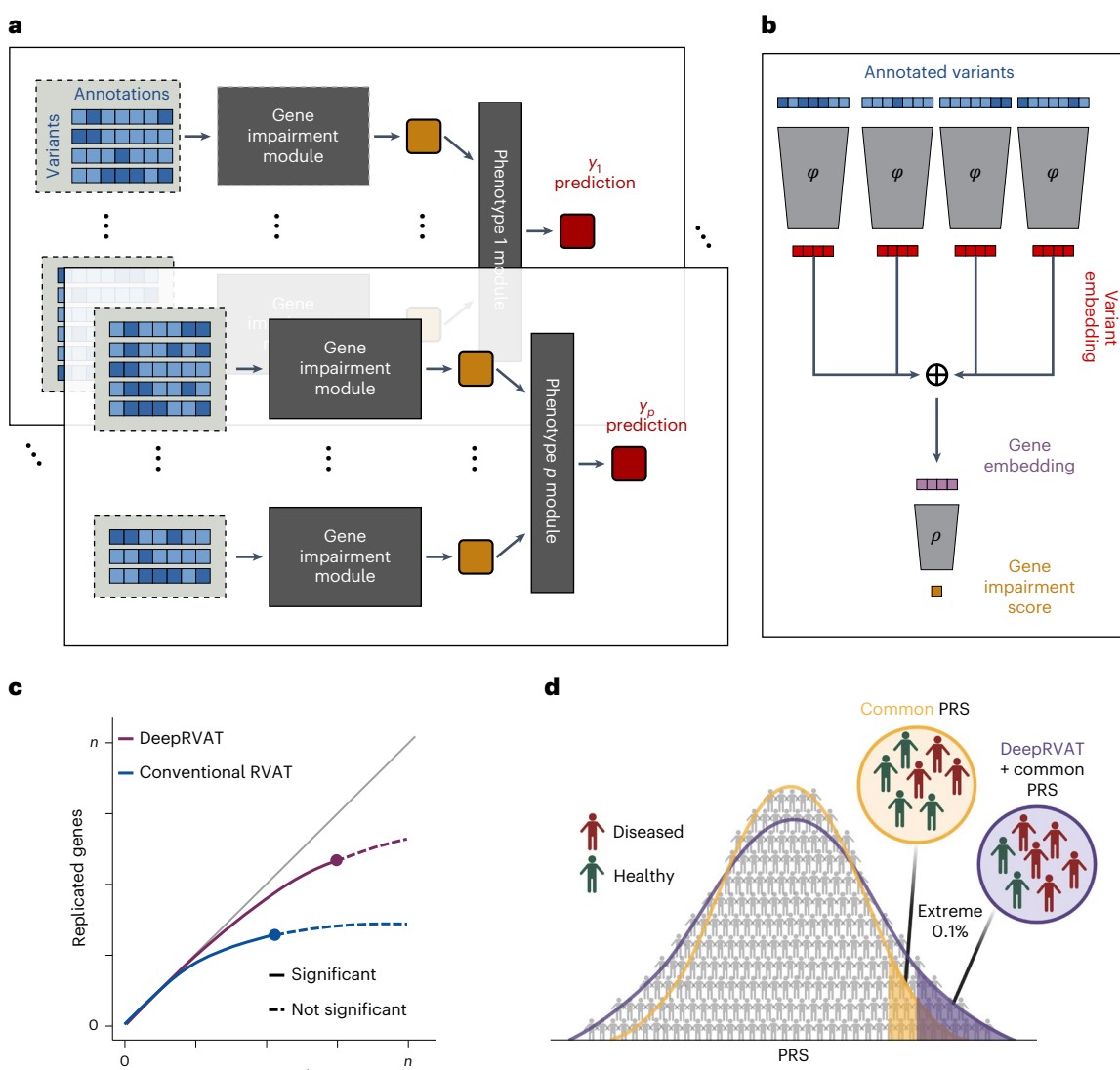

**Fig. 1 | DeepRVAT model overview and downstream use cases. a,b,** DeepRVAT is trained as an end-to-end genotype-to-phenotype model (**a**), using a trait and gene-agnostic gene impairment module (**b**) to infer a scoring function that estimates gene impairment from rare variants and their annotations. The estimated gene impairment scores (orange boxes) are in turn used to explain variation in the set of training traits ($y_1, \dots, y_p$), using trait-specific linear phenotype modules. **b,** The gene impairment module is a set neural network. Annotated variants are fed through an embedding network, $\varphi$, to compute a variant embedding, followed by permutation-invariant aggregation to yield a gene embedding. A second network, $\rho$, estimates the gene impairment score. **c,d,** The trained DeepRVAT gene impairment module is used to estimate impairment scores across genes and individuals, enabling different downstream analyses. **c,** Gene discovery. The impairment scores can be used as input for association tests to conduct gene discovery, assessed by the number of discoveries and replication in held-out data. **d,** Polygenic risk prediction incorporating rare variants. The impairment scores can be used as input for genotype-to-phenotype prediction to, for example, improve risk stratification based on common-variant PRS. The image in **d** is created with BioRender.com.

Initially, we considered 21 quantitative traits of various categories (Supplementary Tables 4 and 5; Methods) to train DeepRVAT, followed by genome-wide association testing of the same traits. Across all traits, DeepRVAT identified 272 gene–trait associations (family-wise error rate (FWER) < 5%; Supplementary Table 6), which corresponds to a 75% increase compared to the widely used baseline approach combining burden and the sequence kernel association test (SKAT), a 19% and 6% increase compared to two existing methods integrating multiple annotations, that of Monti et al.[30] and STAAR[28], respectively (Fig. 2a). We confirmed the statistical calibration of DeepRVAT (Fig. 2b), its robustness to the inclusion of nonassociated seed genes (Extended Data Fig. 2a,b) and verified the expected behavior of the model on synonymous variants (Extended Data Fig. 2c,d).

Next, we evaluated the validity of the discoveries by assessing their replication in at least one of two studies on the full UKBB WES cohort[20,22], which used analysis strategies based on SKAT and burden testing. Notably, across a wide range of nominal significance ranks, the replication rate of DeepRVAT exceeded that of alternative RVAT tests (Fig. 2c). This suggests that not only does DeepRVAT have an improved capacity to detect rare variant associations, but it is also less susceptible to spurious ones.

In assessing the robustness of DeepRVAT to choices in the training procedure, we found that results are stable when downsampling seed genes and training traits (Extended Data Fig. 3a,b) and in the presence of correlated annotations (Extended Data Fig. 3c,d). With the aim of understanding which components of DeepRVAT contribute to its performance, we trained reduced models, considering minimal annotations (MAF, predicted loss of function (pLOF) and missense status), as well as restricting the gene impairment module to linear effects. Both of these simplifications impacted the number of discoveries and the rate

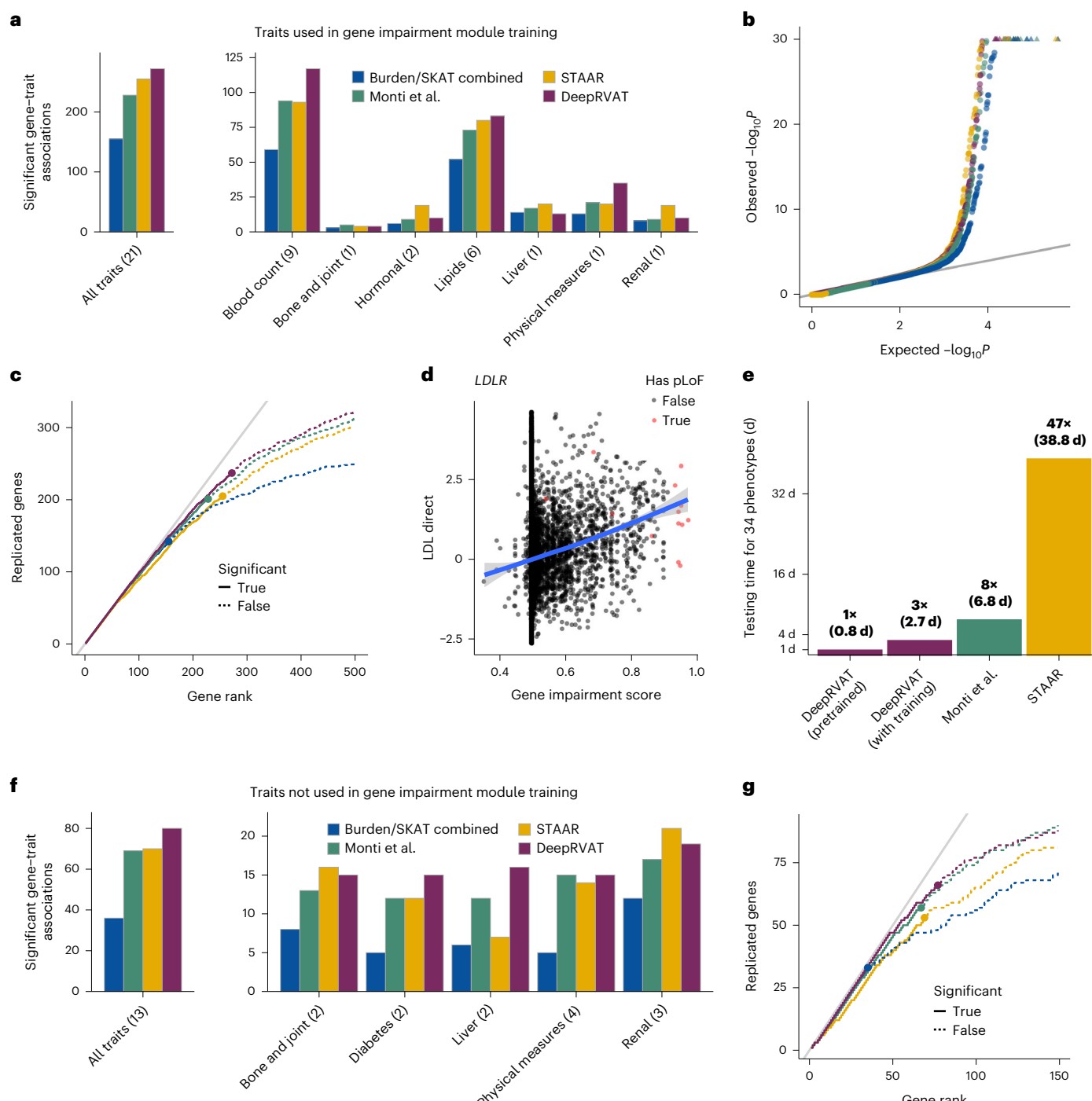

**Fig. 2 | Benchmarking DeepRVAT for gene discovery.** We applied DeepRVAT to gene discovery using WES and 34 traits from 161,822 unrelated UKBB individuals of European ancestry. As alternative methods, we also considered the combination of burden and SKAT tests, each using missense and pLOF variants (burden/SKAT combined), STAAR and the approach from Monti et al.[30]. **a–c**, RVAT analysis on 21 quantitative traits used for training DeepRVAT. **a**, Left: Cumulative number of gene–trait associations (FWER < 5%). Right: Number of gene–trait associations by trait category. **b**, Quantile–quantile (Q–Q) plot of observed versus expected (under the null hypothesis) unadjusted $P$ values across traits. **c**, Replication of cumulative discoveries across traits in larger cohorts (cohort supersets; UKBB full WES release[20,22]; Supplementary Table 7). Shown are, for each method, the number of gene–trait associations that were also discovered in the larger cohort (according to the methodology of the respective studies) versus the rank of their nominal significance. Points indicate the rank

position that corresponds to FWER < 5% (as in **a**). The gray line corresponds to a replication rate of 1. **d**, DeepRVAT gene impairment scores for the *LDLR* gene versus low-density lipoprotein (LDL) cholesterol measurements ($z$-score). Each point represents one individual, with red denoting individuals with at least one pLOF variant in *LDLR*. The blue line denotes the generalized additive model fit, with shaded areas corresponding to 95% confidence intervals. **e**, Empirical computation time when testing for association of 20,000 genes across 34 phenotypes in the UKBB cohort as used in **a**–**d**, **f** and **g**. Multiples are relative to the pretrained DeepRVAT model. Shown are empirical runtimes on a workstation with a 32-core AMD Ryzen Threadripper PRO 5975WX CPU and NVIDIA RTX 4090 24GB GPU. The time for 'DeepRVAT (with training)' includes seed gene discovery. **f**,**g**, Application to 13 quantitative traits not considered during DeepRVAT training. **f** and **g** are analogous to **a** and **c**, respectively.

of replication (Extended Data Fig. 3e,f), indicating that the data-driven scoring function, the ability to consider a larger number of annotations and the capacity to capture nonlinear effects jointly contribute to the overall performance. We also conducted a feature importance analysis for individual annotations (Supplementary Fig. 8), supporting the relevance of a broad spectrum of annotations.

Finally, in addition to enabling well-powered association studies, DeepRVAT gene impairment scores enable fine-grained investigation of the relationship between gene impairment and traits (Fig. 2d and Supplementary Fig. 9). Moreover, DeepRVAT enables computational highly efficient RVAT analyses, in particular when applying the pretrained model (Fig. 2e). To test the universality of the DeepRVAT impairment score, that is, its generalization to new genes and traits, we extended our investigation to 13 additional quantitative traits not considered during training. We found similar benefits in the number of discoveries (Fig. 2f) and their replication (Fig. 2g) as observed for traits used during model training. We also considered conditioning on common-variant effects in the DeepRVAT association test[20,30], confirming that our additional discoveries were not attributable to signals from common variants (Extended Data Fig. 4).

### Improving phenotype prediction by integrating rare variants

The DeepRVAT gene impairment score can be readily leveraged for building phenotype predictors. To demonstrate this, we trained linear regression models that predict phenotypes from DeepRVAT gene impairment scores and public PRSs derived from common variants[35]. For this experiment, we considered the 379,783 unrelated European ancestry individuals from the full UKBB WES release (July 2022, 470k WES release[20]; Methods). Regression models were fitted on the subset of individuals that had been considered for training DeepRVAT and were evaluated on the remaining held-out individuals. We considered the same traits as in the previous section (excluding waist-to-hip ratio (WHR) as it lacked a PRS), using PRS and DeepRVAT impairment scores for genes associated with each trait as features. For comparison, we trained analogous models but with gene-level features derived from single-annotation burdens for trait-associated genes from the burden/SKAT combined method.

DeepRVAT showed the greatest relative improvement in variance explained over common-variant PRS (Fig. 3a and Supplementary Table 8; maximum improvement of 8.26% for alkaline phosphatase). The improvements in variance explained were modest (median improvement of 0.92% for DeepRVAT), which is consistent with common variants explaining most heritability[10]. Nevertheless, it correlated well with burden heritability[10] (Supplementary Fig. 10), indicating that DeepRVAT can capture the overall contribution of rare variants to heritability across different traits. We hypothesized that the importance of rare variants in phenotype prediction would be more apparent when predicting individuals with extreme phenotypes. To assess this, we analogously trained logistic regression models to predict individuals at the extreme phenotypic percentiles. Indeed, the relative improvement was considerably larger for this task (Fig. 3b and Supplementary Table 8; maximum average precision improvement using DeepRVAT of 258.73% for predicting individuals in the lowest percentile of alkaline phosphatase levels). Again, the DeepRVAT impairment score performed significantly better than single-annotation models (Fig. 3b). Ablation analyses showed that the added value of DeepRVAT arises from the combination of a refined gene impairment score and the ability to identify more informative trait-associated genes used as features (Supplementary Fig. 11).

Next, we focused on those individuals with the strongest deviations between the rare variant model and the PRS. Particularly with the DeepRVAT-based model, individuals showing the greatest deviation in predicted phenotypic values were more likely to lie in extreme percentiles (bottom or top 1%; Fig. 3c). Similarly, individuals with extreme phenotype predictions from the DeepRVAT model were more prominently enriched for extreme phenotype measurements compared to alternative rare variant predictors (Fig. 3d). At the most extreme phenotype threshold ($z$-score $\geq 2.30$), DeepRVAT showed a 3.61% greater enrichment of phenotypic outliers in its 99th percentile predictions compared to the common-variant-based PRS model, outperforming the most competitive single-annotation rare variant model, which used AlphaMissense and gave only a 1.9% improvement. These findings demonstrate the advantage of DeepRVAT gene impairment scores in phenotype prediction, particularly for identifying individuals with extreme phenotypes, compared to conventional PRS based on common variants and alternative rare variant models.

### Robustness to imbalanced binary traits and relatedness

Having demonstrated the advantages of DeepRVAT for gene discovery and phenotype prediction, we set out to test the robustness and benefits of the model in practical settings involving related individuals, population structure or low-prevalence case–control designs. To address this, we combined DeepRVAT impairment scores with REGENIE[33], a regression-based association testing framework that is designed to address the three aforementioned challenges. Technically, we conducted single-marker association testing using REGENIE, providing one pseudovariant per gene with the DeepRVAT gene impairment score as a pseudodosage (Methods).

We first assessed the robustness of this DeepRVAT + REGENIE combination to related individuals and population structure for the same quantitative traits as in Fig. 2, however, additionally considering related and multi-ancestry individuals contained in the 200k UKBB WES release. REGENIE conferred robustness when including related individuals, and leveraging multiple ancestries (that is, the full 200k release) led to a net increase in the number of discoveries (Fig. 4a and Extended Data Fig. 5a). Expanding our analyses in a different direction, we returned to unrelated individuals of European ancestry but performed RVAT for 63 binary traits, covering a broad range of disease traits, including heart failure, cataract and pneumonia (Supplementary Table 4). We compared DeepRVAT + REGENIE to the native RVAT implementation of REGENIE (burden and SKAT tests with pLOF and missense masks, variant weights given by Beta(1, 25)-transformed MAF values; Methods), to DeepRVAT without REGENIE, as well as to STAAR and the method of Monti et al.[30]. Strikingly, the latter three were poorly calibrated for these binary traits (Fig. 4b and Extended Data Fig. 5b) and showed an excessive number of gene discoveries for less prevalent conditions (Fig. 4c and Extended Data Fig. 5c), consistent with previous reports[44]. This left only the REGENIE default test and the DeepRVAT + REGENIE combination with calibrated $P$ values (Fig. 4b). These observations demonstrate that DeepRVAT benefits from the Firth penalized logistic regression used within REGENIE, which is robust in rare event settings. Furthermore, we found more than twice as many gene discoveries using REGENIE based on DeepRVAT scores compared to its default burden/SKAT test (Fig. 4d), consistent with the relative power increase observed for quantitative traits.

Finally, we extended our analyses (both conditioned and not conditioned on common variants) to all 469,382 UKBB individuals with available WES, yielding a total of 1,153 significant associations across 97 traits (FWER < 5%; Supplementary Table 9). Remarkably, our analysis yielded 88 associations among the 63 binary traits, 43 of which had not previously been identified in rare variant analyses in the UKBB (Table 1; Methods).

An example of particular interest is a previously unreported association between additional sex combs like 1 (*ASXL1*) and heart failure. *ASXL1* is one of the top genes affected by clonal hematopoiesis, namely clonal expansion of peripheral blood cells carrying somatic mutations but without overt hematological malignancy[45]. Clonal hematopoiesis mutations were reported to increase cardiovascular risk, including heart failure[46]. In particular, somatic mutations of *ASXL1* have been shown to be associated with heart failure and reduced left ventricular

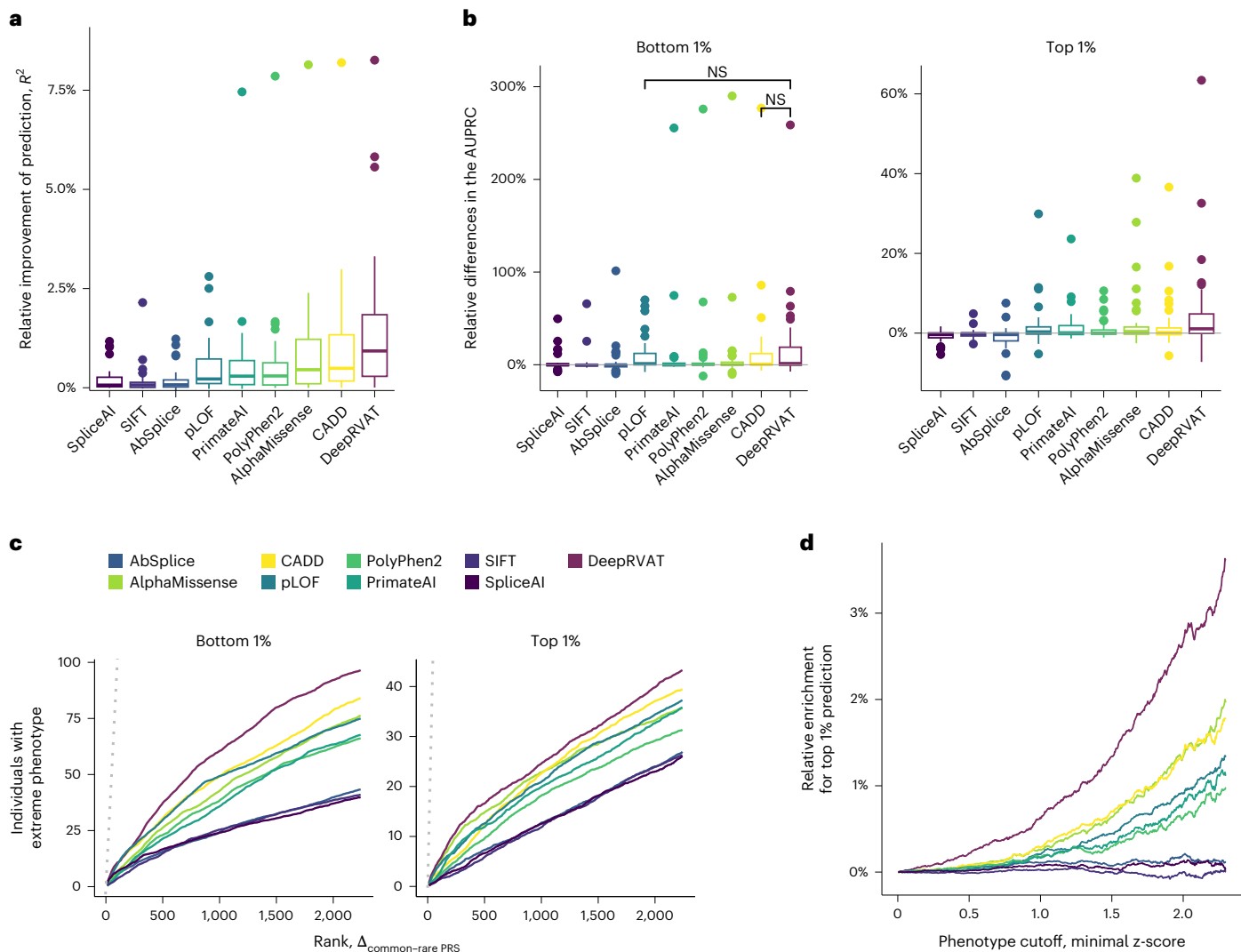

**Fig. 3 | Application of DeepRVAT to phenotype prediction.** We used DeepRVAT gene impairment scores and alternative rare variant gene burdens as features for phenotype prediction. Compared were common variant PRS versus the combination of PRS with either DeepRVAT or one of eight burdens derived from a single annotation. Models were trained on 33 UKBB traits ($n = 154,966$) and evaluated on a held-out fraction of the cohort ($n = 224,817$). **a**, Distribution of relative improvement of prediction performance (coefficient of determination, $R^2$) when including rare variant gene burdens in a linear regression model versus a common variant PRS-only model. The centerline indicates the median; box limits indicate the first and third quartiles; whiskers span all data within 1.5× interquartile ranges of the lower and upper quartiles; values beyond the whiskers are depicted as points. **b**, Analogous comparison as **a** considering logistic regression to stratify individuals in the bottom 1% (left) or top 1% (right) of the phenotypic distribution. Shown are relative differences in the AUPRC between a

model including rare variants versus a common variant PRS-only model. Unless indicated not significant (NS), the relative gains of including the DeepRVAT burden compared to alternative methods as in **a** and **b** are significant ($P < 0.05$, paired one-sided Wilcoxon test). **c**, Rank-based enrichment of individuals with extreme-value phenotypes (top 1% (right) or bottom 1% (left)) among individuals with strongly deviating predictions when using a model that combines PRS and a rare variant burden. Shown is the number of individuals with extreme-value phenotypes when ranked by the magnitude of deviation using a model that includes rare variants versus a common-variant PRS model, averaged across traits. The dotted line corresponds to a perfect ranking. **d**, Enrichment of top 1% phenotype predictions among individuals with extreme phenotypes (exceeding a certain $z$-score cutoff) using a model that includes rare variants versus a common variant PRS model, averaged across traits.

ejection fraction both in humans and mice[47,48]. Our results suggest that germline variants in *ASXL1*, including unreported ones (Extended Data Fig. 6), also predispose to heart failure. Alternatively, but not exclusively, somatic mutations present in the UKBB genotypes may lead to this and possibly further associations. Other notable findings include the association between *HSF4* and the occurrence of cataracts as well as between coronary artery disease and *HHIPL1*. *HSF4* has been implicated in cataract formation in specific canine breeds[49] and through a targeted sequencing study of isolated autosomal-dominant lamellar cataracts in a five-generation British family[50], but not previously from GWAS or RVAT evidence. A GWAS using the FinnGen dataset has reported a low-frequency variant (MAF = 0.7%) in *HHIPL1* associated with coronary

revascularization[51]. Our study strengthens these findings by revealing rare variants in *HHIPL1* associated with coronary artery disease. *HHIPL1* encodes a secreted proatherogenic protein, hedgehog interacting protein-like 1, which regulates smooth muscle cell proliferation and migration[52].

## Discussion

We have introduced DeepRVAT, a data-driven model for association testing and phenotype prediction based on rare variants. Unlike existing methods, DeepRVAT infers the relevance of different annotations and their combinations directly from data. In so doing, DeepRVAT eliminates the need for post hoc aggregation of test results derived

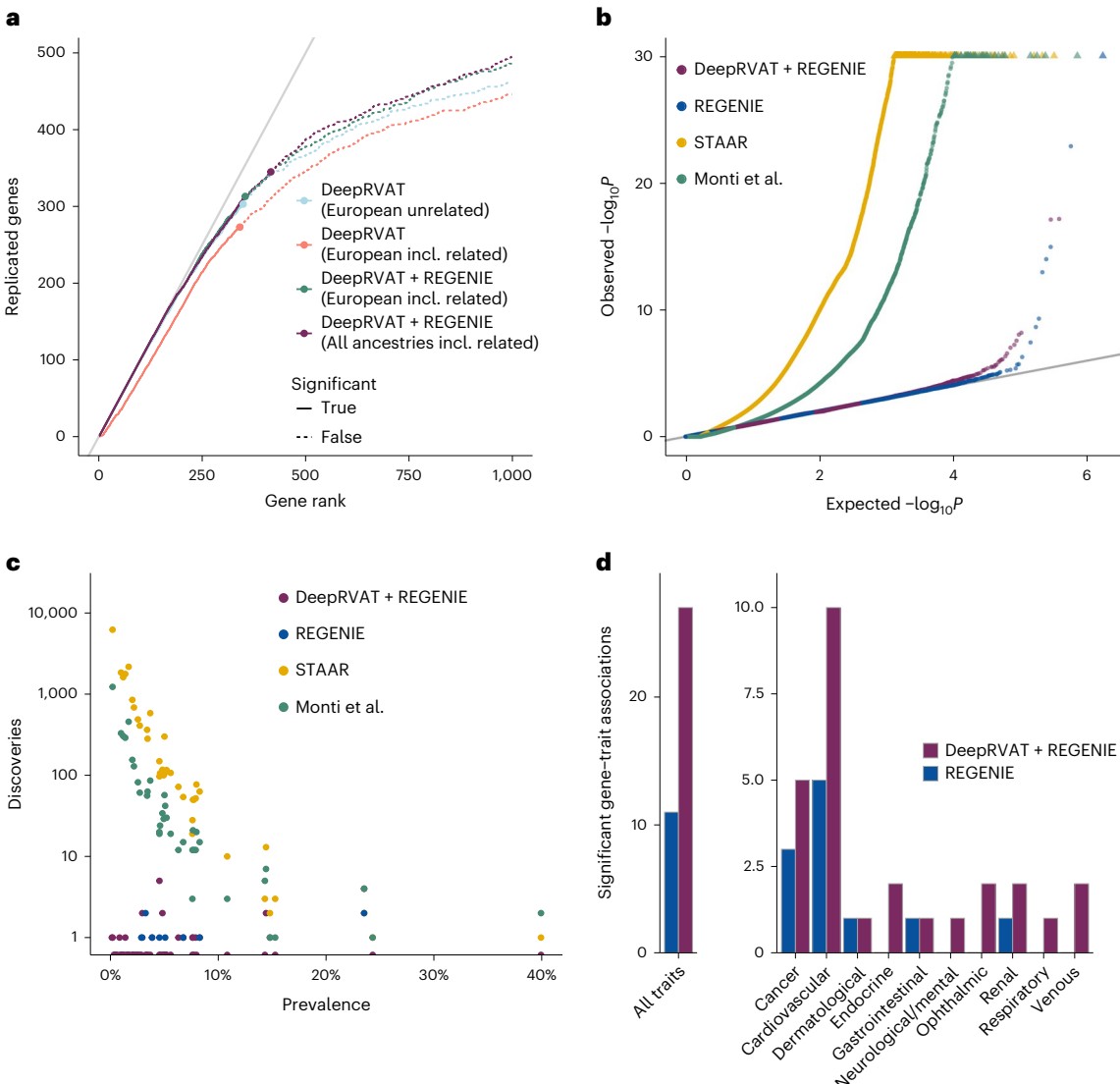

**Fig. 4 | Application of DeepRVAT to related/multi-ancestry individuals and imbalanced binary traits.** To improve robustness to relatedness, population structure and imbalanced binary traits, we combined DeepRVAT with REGENIE (DeepRVAT + REGENIE), using DeepRVAT gene impairment scores as pseudodosages. **a**, Discoveries and replication across 34 quantitative traits, analogous to Fig. 2c,g, considering either the standard DeepRVAT or DeepRVAT + REGENIE. Shown are results obtained on different subcohorts of the UKBB 200k WES release—unrelated individuals of European ancestry (European unrelated), all individuals of European ancestry (European incl. related) and all 200,583 individuals (all ancestries incl. related). **b**–**d**, Application of alternative methods to 63 binary traits, considering unrelated individuals of European ancestry from the UKBB 200k WES release. **b**, Q–Q plots of expected versus observed unadjusted *P* values across all traits. Shown are results from REGENIE with default settings, DeepRVAT + REGENIE, the method of Monti et al.[30] and STAAR. **c**, Scatter plot of the number of gene–trait associations discovered by alternative methods (FWER < 5%) versus trait prevalence. **d**, Cumulative number of gene–trait associations discovered by REGENIE and DeepRVAT + REGENIE (FWER < 5%). Results from the method of Monti et al.[30] and STAAR are not shown given their poorly calibrated test statistics (**b**) and inflated number of discoveries (**c**) on these binary traits.

from individual annotations using multiple-testing schemes. DeepRVAT leverages the flexibility of deep neural networks to integrate rare variant annotations while offering a calibrated statistical framework for gene–trait association testing. DeepRVAT significantly outperforms state-of-the-art methods in gene discovery for 97 traits from the UKBB, leading to a substantial increase in retrieving gene–trait associations with higher replication rates in held-out data.

DeepRVAT represents a conceptual advance by separating trait-agnostic gene impairment scoring on the one hand from gene–trait association testing on the other hand. We have demonstrated the utility of this impairment score for rapid gene–trait association testing by considering traits that were not seen by the model during training. This modular architecture also allows performing gene–trait association testing with dedicated, robust algorithms. Specifically, we

have combined DeepRVAT with REGENIE, enabling robust testing for imbalanced binary traits. We found that even on the well-studied UKBB 470k WES dataset, previously unknown gene–trait associations could be discovered by DeepRVAT.

A second opportunity provided by the impairment score is to estimate the genetic predisposition of individuals by accounting for variants from the full frequency spectrum. We have demonstrated this by combining DeepRVAT with PRS based on common variants, finding considerable benefits over both PRS and conventional burden scores based on single annotations, particularly for extreme phenotypes. DeepRVAT is provided as a user-friendly software package that supports both de novo training of gene impairment modules and the application of pretrained ones, each with substantial improvements in computational efficiency over existing methods.

**Table 1 | Significant gene–trait associations (FWER < 0.05) not found in previous rare variant association studies on UKBB WES[20–22]**

| Trait | Gene | Unconditional analysis | | Conditional analysis | |
|---|---|---|---|---|---|
| | | β | Nominal P value | β | Nominal P value |
| AV or bundle branch block | LMNA | −0.99 | $5.03×10^{-7}$ | −0.99 | $5.00×10^{-7}$ |
| Asthma | LZTS2 | 0.542 | $2.41×10^{-6}$ | 0.557 | $1.47×10^{-6}$ |
| Atrial fibrillation | OR4C11 | −2.07 | $1.75×10^{-6}$ | −2.04 | $2.63×10^{-6}$ |
| | PKP2 | −0.678 | $1.68×10^{-7}$ | −0.7 | $9.21×10^{-8}$ |
| | LMNA | −0.723 | $1.40×10^{-6}$ | −0.763 | $4.64×10^{-7}$ |
| Bradyarrhythmia | LMNA | −1.11 | $8.32×10^{-11}$ | −1.11 | $8.34×10^{-11}$ |
| Breast cancer | NTSR1 | −0.801 | $1.43×10^{-6}$ | −0.806 | $1.50×10^{-6}$ |
| Cataract | HSF4 | −0.612 | $7.04×10^{-7}$ | −0.615 | $6.53×10^{-7}$ |
| | CPAMD8 | −0.332 | $5.96×10^{-9}$ | −0.338 | $3.67×10^{-9}$ |
| | FOXE3 | −0.837 | $4.89×10^{-7}$ | −0.84 | $4.75×10^{-7}$ |
| | BFSP1 | −1.02 | $1.53×10^{-14}$ | −1.03 | $1.01×10^{-14}$ |
| Cholelithiasis | SLC10A2 | −0.591 | $1.22×10^{-13}$ | −0.608 | $3.24×10^{-14}$ |
| | SEL1L | −0.658 | $6.09×10^{-7}$ | −0.67 | $4.31×10^{-7}$ |
| Coronary artery disease | HHIPL1 | −0.532 | $1.42×10^{-6}$ | −0.52 | $2.58×10^{-6}$ |
| Diverticular disease | ADAMTS8 | 0.427 | $5.98×10^{-7}$ | 0.437 | $3.53×10^{-7}$ |
| | COLQ | −0.408 | $4.14×10^{-7}$ | −0.413 | $3.67×10^{-7}$ |
| Glaucoma | FKBP9 | −0.776 | $2.97×10^{-7}$ | −0.774 | $3.50×10^{-7}$ |
| Heart failure | ASXL1 | −0.54 | $3.54×10^{-7}$ | −0.533 | $4.77×10^{-7}$ |
| Hypercholesterolemia | HBB | 0.804 | $8.71×10^{-8}$ | 0.832 | $3.45×10^{-8}$ |
| | APOA5 | −0.519 | $1.02×10^{-7}$ | −0.503 | $3.14×10^{-7}$ |
| | ABCA6 | −0.305 | $2.05×10^{-7}$ | −0.309 | $1.68×10^{-7}$ |
| | NPC1L1 | 0.421 | $5.85×10^{-10}$ | 0.434 | $2.47×10^{-10}$ |
| | ABCA1 | 0.3 | $2.64×10^{-8}$ | 0.305 | $1.92×10^{-8}$ |
| Hypertension | PDE3B | 0.257 | $9.40×10^{-7}$ | 0.269 | $4.05×10^{-7}$ |
| | INPPL1 | 0.407 | $4.95×10^{-9}$ | 0.415 | $3.72×10^{-9}$ |
| | BLM | −0.269 | $8.53×10^{-7}$ | −0.281 | $3.84×10^{-7}$ |
| | RMC1 | −0.536 | $1.64×10^{-6}$ | −0.543 | $1.54×10^{-6}$ |
| | NPR1 | −0.436 | $8.47×10^{-11}$ | −0.453 | $2.55×10^{-11}$ |
| | REN | 0.699 | $2.20×10^{-8}$ | 0.722 | $1.06×10^{-8}$ |
| | ENPEP | −0.247 | $4.93×10^{-9}$ | −0.242 | $1.55×10^{-8}$ |
| | GUCY1A1 | −0.526 | $8.85×10^{-10}$ | −0.524 | $1.59×10^{-9}$ |
| Hypertrophic cardiomyopathy | MYH7 | −2.78 | $3.19×10^{-16}$ | −2.78 | $3.19×10^{-16}$ |
| Irritable bowel syndrome | PRMT8 | −1.71 | $4.98×10^{-7}$ | – | – |
| Osteoarthritis | CSPG4 | 0.295 | $1.85×10^{-6}$ | 0.298 | $1.50×10^{-6}$ |
| Parkinson's disease | CFDP1 | −1.77 | $2.37×10^{-6}$ | −1.77 | $2.19×10^{-6}$ |
| Pneumonia | TET2 | −0.554 | $4.82×10^{-7}$ | −0.553 | $5.05×10^{-7}$ |
| Prostate cancer | SAMHD1 | −1.09 | $8.68×10^{-8}$ | −1.11 | $1.01×10^{-7}$ |
| Skin cancer | TYR | −0.534 | $9.78×10^{-8}$ | −0.62 | $1.02×10^{-9}$ |
| | SLC45A2 | −0.464 | $4.76×10^{-7}$ | −0.486 | $1.64×10^{-7}$ |
| | CDKN2A | −2.11 | $1.41×10^{-13}$ | −2.13 | $1.32×10^{-13}$ |
| Venous thromboembolism | STAB2 | −0.446 | $8.97×10^{-9}$ | −0.445 | $1.07×10^{-8}$ |
| | PROC | −2.38 | $7.81×10^{-30}$ | −2.39 | $4.54×10^{-30}$ |
| | F11 | 0.801 | $7.36×10^{-7}$ | 0.799 | $8.34×10^{-7}$ |

Unadjusted P values as returned by REGENIE are listed. Conditional analyses were performed for all traits with at least one GWAS signal at $P < 10^{-7}$, which applied to all traits but irritable bowel syndrome, myocardial infarction and colorectal cancer.

Although we found that DeepRVAT advances the state-of-the-art in two use cases central to genetics, the model is not free of limitations. First, while we have shown that DeepRVAT can cope with a potentially very large number of annotations, including correlated ones, the choice of informative annotations remains empirical. Furthermore, generating the annotations for all variants, which is a prerequisite to training DeepRVAT, can be computationally expensive, in particular if these are based on massive deep neural networks. Second, we benchmarked DeepRVAT using exome-sequencing data, but whole-genome sequencing promises valuable insights into noncoding regions with numerous rare variants of uncertain impact. We hypothesize that the added value of DeepRVAT, which incorporates rich variant annotations, might be even larger in this context. Finally, while we showed that DeepRVAT is robust to various rare variant frequency cutoffs, it is still based on a dichotomy between rare and common variants. Recent reports have shown substantial overlap between GWAS loci and rare variant associations[10,11,20]. These insights suggest the potential for future developments to jointly model and estimate rare and common variant effects in a unified framework.

The analysis of rare variants will remain a major topic in quantitative genetics modeling. DeepRVAT belongs to the class of approaches that model gene impairment, a strategy that underpins high-impact variant filters in burden tests[20,22,23] or protein function impairment[25]. Our contribution is to better optimize the impairment score by learning a more flexible model directly from annotated variants on cohort data. Notably, we found this gene impairment score to generalize well across traits. This transferability, combined with DeepRVAT's sensitivity and computational efficiency, is an important feature that will facilitate its application to study allelic series[53], to conduct rapid rare variant PheWAS and to discover rare variant associations in smaller cohorts from case–control studies.

## Online content

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

## Methods

A more detailed exposition of the methods may be found in the Supplementary Methods.

### Ethics statement

UKBB protocols are overseen by the UKBB Ethics Advisory Committee. Informed consent was obtained for all participants. Participants who revoked consent were removed from the analysis. The original approval for the UKBB was granted in 2011 by the National Research Ethics Service Committee North West−Haydock. The approval was renewed in 2016 and 2021 by the Health Research Authority, North West−Haydock Research Ethics Committee. This research has been conducted using the UKBB resource under applications 25214, 44108 and 81358.

### DeepRVAT model

**Model and applications.** The DeepRVAT gene impairment module is trained as part of an end-to-end multiphenotype prediction model. The pretrained gene impairment module can be leveraged in various downstream tasks, including RVAT and augmenting PRS with rare variant effects.

**Input data.** The input data for DeepRVAT consists of unordered sets of variants. The variant set for individual $i$ and gene $j$ is defined as follows:

$$V_{ij} = \left\{ (a_{kl})_{l=1,\dots,d} \mid \text{variant } k \text{ present in individual } i, \text{ gene } j \right\},$$

where $a_{kl}$ is the $l$th annotation of variant $k$.

**Model architecture.** We used a deep set architecture[34]. The variant set is first passed through a submodule $\varphi$, which computes a variant embedding $\varphi(x_k)$ for each $x_k = (a_{k1}, \dots, a_{kd}) \in V_{ij}$. Next, a gene embedding is computed using a fixed and permutation-invariant aggregation function $f$ (we used the element-wise maximum), and then a second submodule $\rho$ computes a scalar gene impairment score. Finally, we pass the result through a sigmoid function. In full,

$$\psi(V_{ij}) = \sigma\left( \rho\left( f\left( \{\varphi(x_k)\}_{x_k \in V_{ij}} \right) \right) \right),$$

where both $\varphi$ and $\rho$ are multilayer perceptrons (MLPs), and $\sigma$ is the sigmoid (logistic) function.

**Seed genes.** Training of the gene impairment module begins by selecting, for each training phenotype, a set of trait-specific 'seed genes' from the set of all protein-coding genes. In this study, we base these on the results of alternative RVAT methods, specifically the 'burden/SKAT combined' method described below.

**Training objective.** DeepRVAT is trained in a multitask framework across multiple phenotypes. We estimate the $p$th phenotype $y_i^{(p)}$ for individual $i$ as a linear combination of the gene impairment scores and covariates:

$$\hat{y}_i^{(p)} = \mathbf{x}_i^T \alpha^{(p)} + \sum_{j \in S^{(p)}} w_j^{(p)} \psi(V_{ij}),$$

where $\mathbf{x}_i$ is the vector of covariates for individual $i$ and $S^{(p)}$ is the set of seed genes for phenotype $p$. The parameters of $\psi$ are shared across all variants, genes and phenotypes, while $\alpha^{(p)}$ and $w_j^{(p)}$ are phenotype specific. The loss is given by the mean across phenotypes,

$$L(y_i, \hat{y}_i) = \frac{1}{P} \sum_{p=1}^{P} L\left( y_i^{(p)}, \hat{y}_i^{(p)} \right).$$

**Training-validation split.** A data point for the DeepRVAT multiphenotype model was given by an individual−phenotype pair. Before training, data points were shuffled, and a validation set consisting of 20% of individuals was selected at random for each phenotype.

**Hyperparameters and software.** For the variant embedding $\varphi$, we used a two-layer MLP with a width of 20. We used an MLP with two hidden layers of width 10 for $\rho$. In both networks, leaky rectified linear units with a negative slope of 0.01 were used as the activation functions. We used the mean-squared error (MSE) loss and the AdamW optimizer with a learning rate of 0.001. The batch size was 1,024. Early stopping was used to select the checkpoint with the lowest validation MSE loss. Training proceeded for a minimum of 50 and a maximum of 1,000 epochs. All DeepRVAT models were implemented in PyTorch (v1.13.1) and PyTorch-Lightning (v1.5.10).

**Ensembling and CV scheme.** The dataset is first used for seed gene discovery using the 'burden/SKAT combined' method described below. Next, we partition the dataset across samples into $K$ equally sized subsets (in this study, we used $K = 5$). We hold out one subset and train on all others, followed by computing gene impairment scores on the held-out samples using the trained model. This process is repeated for all subsets, resulting in a set of $K$ models that can be used to estimate gene impairment scores for association testing, thereby avoiding information leakage without compromising sample size. Training within each CV fold is repeated for six random restarts, resulting in an ensemble of models. When estimating burden scores, all folds and random restarts that did not include the individual during training are averaged.

**Integration into single-marker association testing frameworks.** Because DeepRVAT provides a single score per gene and sample, it can be seamlessly integrated into any tool that carries out single-marker association testing with genotype dosages. Practically, we implement this by providing a script that uses the bgen package (v1.6.1) to convert the (samples × genes) matrix of DeepRVAT scores to a BGEN file of pseudovariants. The DeepRVAT gene impairment score, $\psi(V_{ij})$ is stored as the probability $p_{ij} = (\psi(V_{ij}), 0, 1 - \psi(V_{ij}))$ of homozygous alternate, heterozygous and homozygous reference alleles so that the pseudo-dosage $d_{ij} = 2\psi(V_{ij})$ lies in the usual range of [0,2].

### Alternative RVAT methods

**Burden tests and SKAT.** We implemented burden and SKAT tests following ref. 22, using the score test from the SEAK package[30] (v0.4.3). All combinations of burden and SKAT tests restricted to either pLOF or missense variants were carried out. We also created a combination test (burden/SKAT combined) using the full set of $P$ values from all four individual tests. Variants were weighted with the betadensity function, that is, Beta(MAF; 1, 25). Ensembl VEP[37] was used to annotate missense and pLOF variants, with the latter composed of stop gained, start lost, splice donor, splice acceptor, stop lost or frameshift variants. In SKAT tests, variants with minor allele count (MAC) ≤10 were collapsed as described in ref. 44. Due to computational constraints, we skipped genes with over 5,000 markers, impacting only one gene (Titin) for missense variant tests.

**STAAR.** STAAR tests were implemented using the STAAR package (v0.9.6) provided by the authors and following its vignette as well as the original publication[28]. To ensure optimal comparability with DeepRVAT, the same annotations as for DeepRVAT (described below) were used. As required for the STAAR procedure, each annotation was PHRED-scaled. Following ref. 28, STAAR $P$ values were computed for five variant groups, namely (1) putative loss-of-function (stop gain, stop loss and splice), (2) missense, (3) disruptive missense, (4) putative loss-of-function and disruptive missense and (5) synonymous variants. We defined disruptive missense variants to be those that were predicted to be both 'deleterious' by SIFT[14] and 'probably damaging' by PolyPhen2[16].

**Method by Monti et al..** We used the same annotations, variant weight thresholds and variant kernel architectures as outlined in ref. 30. Annotation scores were obtained according to the details provided in Supplementary Table 3. We used the score test from SEAK to compute *P* values. In the case of missense and splicing tests, if either the collapsing or kernel-based association test yielded nominal significance (*P* < 0.01), we performed joint testing with pLOF variants. The *P* values from these tests were aggregated using the Cauchy combination method.

**REGENIE.** Burden and SKAT tests were run using both missense and pLOF masks, yielding four combinations. For burden tests, we used the default REGENIE strategy of collapsing variants to gene level using the maximum number of alternate alleles across sites. We used the approximate Firth likelihood ratio test for *P* values less than 0.01. Identically to the tests implemented in SEAK mentioned above, weights for SKAT tests were computed as Beta(MAF;1,25).

### Combination of multiple *P* values per gene and multiple-testing correction
The burden/SKAT combined method, the method Monti et al.[30] and STAAR each yielded multiple *P* values per gene. These were aggregated at the gene level using the Bonferroni procedure, and the smallest *P* value per gene was retained. For all methods, we again used Bonferroni correction across all 19,388 tested genes.

### Expected allele frequency (EAF) filtering
Following ref. 22, we restricted testing to genes that passed an EAF filter of at least 50. The EAF is defined as CAF × *n*, where CAF is the cumulative allele frequency (the sum of allele frequencies of all qualifying variants *j* in the gene) and *n* is the cohort size for quantitative traits or the number of cases for binary traits.

### UKBB WES data
**Exome-sequencing data.** WES (+100 bp overhang) was performed on 469,779 participants from the UKBB[20], for which the methods have been described in the earlier release of data from approximately 50,000 individuals[54]. We refer to this as the UKBB 470k WES dataset and use the UKBB 200k WES dataset to refer to the interim release from 200,633 participants.

**Variant data and quality control (QC).** For both cohorts mentioned above, variant calling data were downloaded from the UKBB as project-level VCF (pVCF) files. We applied additional QC following ref. 54 and using bcftools[55] (v1.10.2)—minimum read depth of 7 for SNPs and 10 for InDels; at least one homozygous variant genotype or at least one sample per site with an allelic balance ratio greater than 15% for SNPs and 10% for InDels; fraction of missing genotypes <10%; and Hardy–Weinberg equilibrium *P* value > $10^{-15}$. Additionally, we filtered out individuals with >10% missing genotype rate and those who had withdrawn from the study. Finally, following the analysis best practices recommended by UKBB, we applied a coverage filter, requiring that at least 90% of all genotypes for a given variant have a read depth of at least 10. This resulted in datasets with 200,583 individuals and 12,704,497 variants (UKBB 200k WES), and 469,382 individuals and 26,141,967 variants (UKBB 470k WES).

**Custom sparse genotype data format.** To enable fast, multiple iterations over the WES datasets, we created a custom sparse dataset in Hierarchical Data Format 5 (HDF5 v1.10.6). Details are provided in the Supplementary Methods. For UKBB 200k WES, the HDF5 dataset had a storage size of approximately 100 GB, compared to multiple terabytes for the original pVCF files.

**Covariates.** We retrieved genetic sex, sample age and the first 20 genetic principal components (PCs) directly from UKBB (Supplementary Table 5). We computed $age^2$ and age × sex to use as additional covariates. Covariates were included in association testing with all methods and when training DeepRVAT.

**Variant-to-gene assignments.** Variants were assigned to genes using those protein-coding genes and associated exons marked as golden in the merged Ensembl/HAVANA genome annotations (GENCODE release 38). We assigned a variant to a gene if it was located at most 300 bp from an associated exon.

### Variant annotations
The full collection of variant annotations used and their sources are provided in Supplementary Table 3. Here we give details on processing for those annotations that were not used directly in the form output by the source.

**MAF.** MAF values for variants were first replaced with the maximum of the MAF in the UKBB cohort and in gnomAD release 3.0 (non-Finnish European population). Following ref. 23, the MAF $p_j$ of each variant *j* was then transformed according to the formula $[p_j(1-p_j)]^{-\frac{1}{2}}$ for use in modeling.

**DeepSEA.** To improve model fitting and avoid overfitting, we performed PC analysis and restricted to the first six PCs of DeepSEAs 919 predicted variant effects.

**SpliceAI.** SpliceAI provides four 'delta scores' indicating a variant's predicted effect on cryptic splicing[40]. We computed the maximum of these four scores and used it as a single annotation.

**AbSplice.** We computed the maximum predicted effect across tissues and used this as a single annotation.

**DeepRiPe.** As in ref. 30, we predicted the effects of genetic variants on the binding of six RBPs over three cell lines.

**MAF thresholds.** For DeepRVAT training, we used variants with MAF < 1%. For association testing with all methods, we designated rare variants as having MAF < 0.1%. Additionally, for both training and association testing, we restricted to variants with PHRED-scaled CADD value >5.

### Phenotype data
All phenotype data (Supplementary Table 4) were obtained directly from UKBB, except for WHR, which was computed as the ratio of UKBB data field 48 to data field 49 and corrected for body mass index by regressing out the corresponding data field 21001. Phenotype values were quantile transformed to match their empirical distributions to a standard normal distribution. For individuals with reported statin usage, we adjusted cholesterol (30690) by dividing by 0.8 and LDL-direct (30780) by dividing by 0.7, following refs. 56,57. Statins considered were obtained from ref. 58 and matched to UKBB treatment codes (20003). Binary traits were extracted using the definitions from ref. 21.

### DeepRVAT training and association testing on UKBB data
**Subselected cohorts.** Because the various methods used for benchmarking control for sample relatedness and population structure differently, or not at all, we retained only unrelated individuals of European genetic ancestry from the UKBB 200k WES dataset for DeepRVAT model training and benchmarking against alternative RVAT methods. Individuals related to third degree or closer were identified according to UKBB Resource 668. Individuals of European ancestry were identified using UKBB data field 22006 (termed 'Caucasian'). This filtering resulted in a dataset (denoted UKBB 200k unrelated European ancestry

below) of 161,822 individuals. For testing the integration of DeepRVAT with REGENIE (Fig. 4a, we additionally used all 167,214 individuals of European genetic ancestry.

**Training and gene impairment scoring.** Seed gene discovery and DeepRVAT training were carried out on the UKBB 200k unrelated European ancestry dataset. Training and gene impairment scoring were done according to the CV scheme described above. An ensemble consisting of all 30 models from the CV step (six ensemble models from five training folds) was used to compute gene impairment scores for the remaining 307,560 individuals from the UKBB 470k WES cohort.

**Association testing.** For the method denoted DeepRVAT, association testing was carried out using the score test from SEAK (v0.4.3). Association testing for the method DeepRVAT + REGENIE was carried out with REGENIE (v3.4.1). Following the REGENIE documentation, for step 1, we selected approximately 500k (precisely, 483,446) imputed SNPs from UKBB data field 22828 using the following filtering: MAF < 0.06, MAC > 100, genotyping rate >0.99, Hardy–Weinberg $P$ value ≥ $10^{-15}$ and sample missingness <0.1. Additionally, we pruned SNPs with a pairwise linkage disequilibrium $r^2$ threshold of 0.9, using a window size of 1,000 and a step size of 100. Step 2 of REGENIE was run on DeepRVAT gene impairment scores for each gene, derived as described above. For quantitative traits, the default options of REGENIE were used. For binary traits, we used the approximate Firth likelihood ratio test with a $P$ value threshold of 0.01. To account for multiple testing across genes, we applied Bonferroni correction for DeepRVAT and alternative methods.

### Comparison with other UKBB RVAT studies
We compared our results to gene–trait associations from two studies[20,22] on larger WES cohorts from UKBB (454,787 and 394,841 individuals, respectively). We counted as a discovery any association that was considered significant according to the methodology of the study. To compute replication for quantitative traits, we ranked all gene–trait associations by $P$ value and, at each rank, counted the number of associations that overlapped with discoveries from the larger cohorts.

### Conditional association tests
For associations that were significant after multiple testing correction, we conducted conditional association tests using GWAS summary statistics from the Pan-UKBB study[59]. Independently associated variants were identified from GWAS summary statistics through LD-based clumping using PLINK[60] (v1.9) with default parameters, restricting to associations with a $P$ value < $10^{-7}$ and MAF > 1%. If a binary trait definition used in this study did not exactly match a single GWAS from Pan-UKBB, we combined $P$ values from all relevant GWASs that covered parts of the trait definition before performing clumping. For association testing with SEAK in the method denoted DeepRVAT, independently associated variants within 500 kb around the gene boundaries were incorporated as covariates in the conditional analysis. For association testing with REGENIE (that is, DeepRVAT + REGENIE), all variants independently associated with a specific trait were considered for all genes.

### Phenotype prediction using DeepRVAT and alternative rare variant scores
**Dataset.** Training and evaluation of the regression models were done on two disjoint datasets, both restricted to unrelated individuals of European ancestry. A total of 154,966 (from UKBB 200k WES) and 224,817 individuals (from UKBB 470k WES, not found in UKBB 200k WES) were used for training and evaluation, respectively.

**PRS computation.** Common PRS variants and effect sizes were all obtained from the Polygenic Score Catalog[61] using the study from ref. 35. The catalog numbers of each common variant PRS are listed in Supplementary Table 4.

**Gene discovery.** Only the training individuals were used for gene discovery. For alternative burdens, we used the method 'burden/SKAT combined'. Retaining associations at FWER < 0.05 resulted in a set of genes $G_b^{(p)}$ for phenotype $p$ to use in the baseline prediction models. For DeepRVAT, gene discovery was conducted across all 33 traits of interest exclusively on training samples, following the method outlined above, using gene impairment scores obtained using the CV scheme. This yielded a set of trait-associated genes $G_d^{(p)}$ at FWER < 0.05.

**Burden and gene impairment scores.** On the test set, alternative burdens were computed as the maximum across all variants in a given individual and gene (excluding SIFT, where the minimum was used). DeepRVAT gene impairment scores for the test set were computed analogously to those used in association testing, using all models trained as part of the CV scheme on the evaluation set.

**Phenotype predictor training and evaluation.** For simplicity, we describe models for predicting raw phenotype values. Prediction of extreme values is analogous, with logistic regression on the binary target replacing linear regression.

**Baseline.** As a baseline phenotype predictor, we consider a regression model where the explanatory variables comprise covariates (as described above) and the common variant PRS score:

$$\hat{y}_i^{(p)} = \alpha^T \mathbf{x}_i + \beta_c^{(p)} c_i^{(p)},$$

where $c_i^{(p)}$ is the common variant PRS score of sample $i$ for phenotype $p$ and, as given above, $\mathbf{x}_i$ is the vector of covariates for sample $i$.

**Extension with rare variants.** To incorporate the effects of rare variants into the phenotype predictors, we extended the common variant PRS models by the rare burden scores of significant genes, with models incorporating DeepRVAT or alternative burdens given, respectively, by

$$\hat{y}_i^{(p)} = \alpha^T \mathbf{x}_i + \beta_c^{(p)} c_i^{(p)} + \sum_{j \in G_d^{(p)}} \beta_j^{(p)} \psi_r^* (V_{ij}),$$

$$\hat{y}_i^{(p)} = \alpha^T \mathbf{x}_i + \beta_c^{(p)} c_i^{(p)} + \sum_{j \in G_b^{(p)}} \beta_j^{(p)} s_{ij}.$$

The difference lies in whether DeepRVAT or alternative burdens $s_{ij}$ are used, and additionally the burdens and learned gene weights $\beta_j^{(p)}$ range over either the 'burden/SKAT combined' gene set $G_b^{(p)}$ or the DeepRVAT gene set $G_d^{(p)}$.

**Model fitting.** The linear and logistic regression models were fit in R (v4.2.0) using the functions `lm` and `glm` (respectively) from the stats package using the family `binomial()` for logistic regression models and otherwise retaining the default parameters.

**Phenotype predictor assessment.** We calculated the relative improvement of the model that leverages rare variant burdens compared to the common variant PRS-only model as

$$\text{Relative}\Delta M = \frac{M_{\text{rare}} - M_{\text{PRS}_{\text{only}}}}{M_{\text{PRS}_{\text{only}}}},$$

where $M$ denotes the area under the precision–recall curve (AUPRC) or $R^2$. Next, for each phenotype and individual, we calculated the absolute difference between the predicted phenotype values obtained from a linear model using either the common variant PRS alone or the common variant PRS together with the rare variant burdens and ranked individuals based on the magnitude of this difference. At each rank, we determined the count of individuals exhibiting outlier phenotypes,

specifically those falling within the top or bottom 1% of the phenotypic distribution. Finally, we tested the enrichment of phenotype predictor outliers in individuals with extreme phenotypes. For each $z$-score phenotype outlier cutoff, we identified individuals above the phenotypic cutoff and determined the proportion of these individuals with a predicted phenotype value exceeding the 99% quantile. Enrichment scores were scaled relative to the baseline population ($z$-score = 0) and compared to the common PRS-only model.

## Statistics and reproducibility
No statistical method was used to predetermine the sample size. We did not use any study design that required randomization or blinding. For benchmarking experiments we restricted to unrelated individuals of European ancestry. All individuals (related and multi-ancestry) were included for biological discovery.

## Reporting summary
Further information on research design is available in the Nature Portfolio Reporting Summary linked to this article.

## Data availability
The genetic, phenotype and covariate data are protected and only available to researchers who have valid and approved research applications for these data within the UKBB (www.ukbiobank.ac.uk).

GENCODE release 38 can be downloaded from https://www.gencodegenes.org/human/release_38.html.

PRSs used for the phenotype prediction can be downloaded from https://www.pgscatalog.org/.

GWAS summary statistics used for conditional analysis can be downloaded from https://pan.ukbb.broadinstitute.org.

Significant gene–trait associations used for replication analyses are provided in Supplementary Table 7. These can also be accessed at gs://ukbb-exome-public/500k/results/results.mt for the study in ref. 22 and in the supplementary data from ref. 20.

The association testing results from DeepRVAT + REGENIE on the 470k UKBB WES dataset, covering all genes and traits for all ancestries and European ancestry only, are available on Zenodo (https://doi.org/10.5281/zenodo.12736824) (ref. 62).

## Code availability
UKBB data preprocessing was done using our custom preprocessing pipeline available at https://deeprvat.readthedocs.io/en/latest/preprocessing.html (ref. 63).

The code to run DeepRVAT can be found at https://github.com/PMBio/deeprvat/ (ref. 63).

Pretrained DeepRVAT models can be found at https://github.com/PMBio/deeprvat/tree/main/pretrained_models (ref. 64).

The code to run all analyses done in this paper and regenerate the figures can be found at https://github.com/PMBio/deeprvat-analysis/ (ref. 65).

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

## Acknowledgements
The authors would like to thank M.J. Bonder, T. Bechtler, M. Saraswat, A. Senacheribbe, K. Ueltzhöffer, D. Vagiaki, H. Susak, F.P. Casale, R. Monti and M.F. Clarke for helpful discussions. This research has been conducted using data from UKBB, a major biomedical database (project IDs 25214, 44108 and 81358), and was supported by the Initiative and Networking Fund of the Helmholtz Association. B.C., K.M., M.M., F.M. and M.W. were supported through state funds approved by the State Parliament of Baden-Württemberg for the Innovation Campus Health + Life Science Alliance Heidelberg Mannheim. E.H. was funded in part by the Helmholtz Association under the joint research school Munich School for Data Science—MUDS. This study was supported by the Deutsche Forschungsgemeinschaft (DFG; German Research Foundation; project ID 403584255-TRR 267 to Z.C. and J.G.). F.B. was supported through the German Bundesministerium für Bildung und Forschung (BMBF) through the ERA PerMed project PerMiM (grant 01KU2016B). The German BMBF supported the study through the Model Exchange for Regulatory Genomics project (MERGE; grants 031L0174A and 031L0174C). Figure 1d was created with BioRender.com. The funders had no role in study design, data collection and analysis, decision to publish or preparation of the paper.

## Author contributions
B.C., E.H., J.G. and O.S. conceived the method. F.B. made additional conceptual contributions to the method. B.C., E.H., H.O., M.M., M.W. and F.R.H. prepared the data. B.C., E.H., H.O., K.M., M.M., F.M. and M.W. implemented the methods and analyzed the data. B.C., E.H., J.G. and O.S. interpreted the results and wrote the paper. Z.C. supported

in biological interpretation. J.L. contributed to the conditional analysis.

## Funding

## Competing interests

O.S. is a paid advisor of Insitro. B.C. holds equity in Tachyon Therapeutics. The other authors declare no competing interests.

## Additional information

**Extended data** is available for this paper at https://doi.org/10.1038/s41588-024-01919-z.

**Correspondence and requests for materials** should be addressed to Brian Clarke, Julien Gagneur or Oliver Stegle.

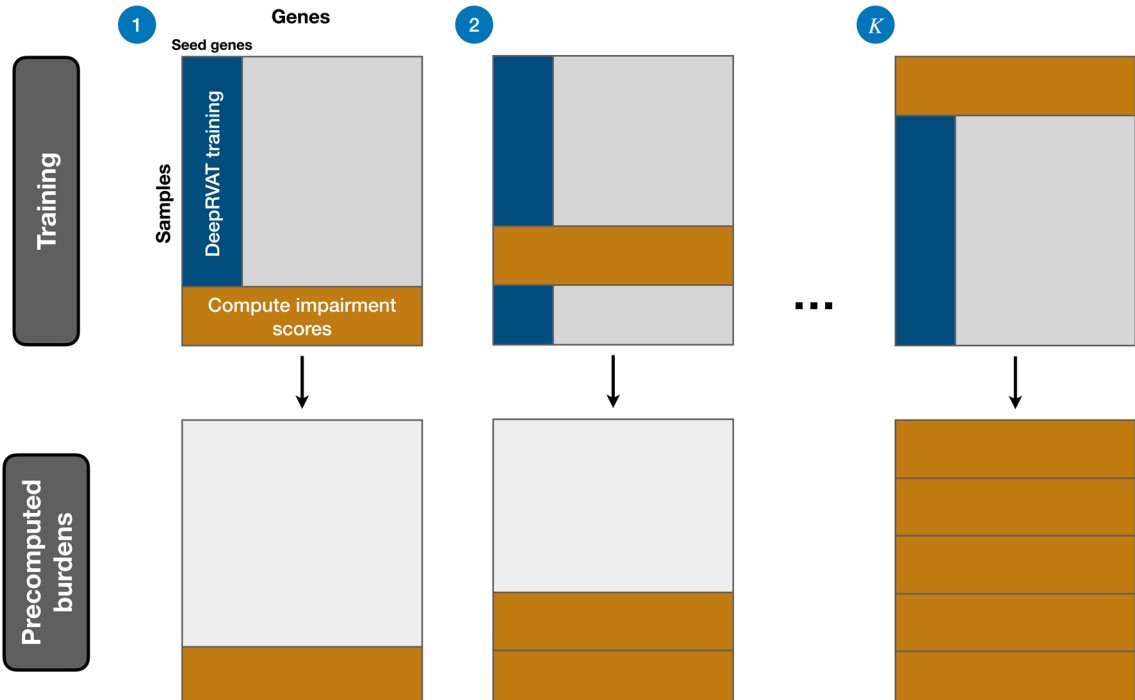

**Extended Data Fig. 1 | Cross-validation training scheme.** The dataset *D* is first utilized for seed gene discovery using the burden/SKAT combined method (not shown). Next, *D* is divided across samples into *K* equally sized subsets. For each fold *k*, six gene impairment modules are fitted using different random initializations on the training fraction (blue) and discovered seed genes. Finally, a gene impairment score is computed for each sample in the test set (orange) and each gene, averaging over the ensemble of all six models. This yields a dataset of the same sample size, which can be used for association testing, while avoiding overfitting by computing gene impairment scores on samples not used during training of the given model. When a pre-trained DeepRVAT model is applied to independent data, the ensemble average score across all gene impairment modules that were not trained using that individual is used.

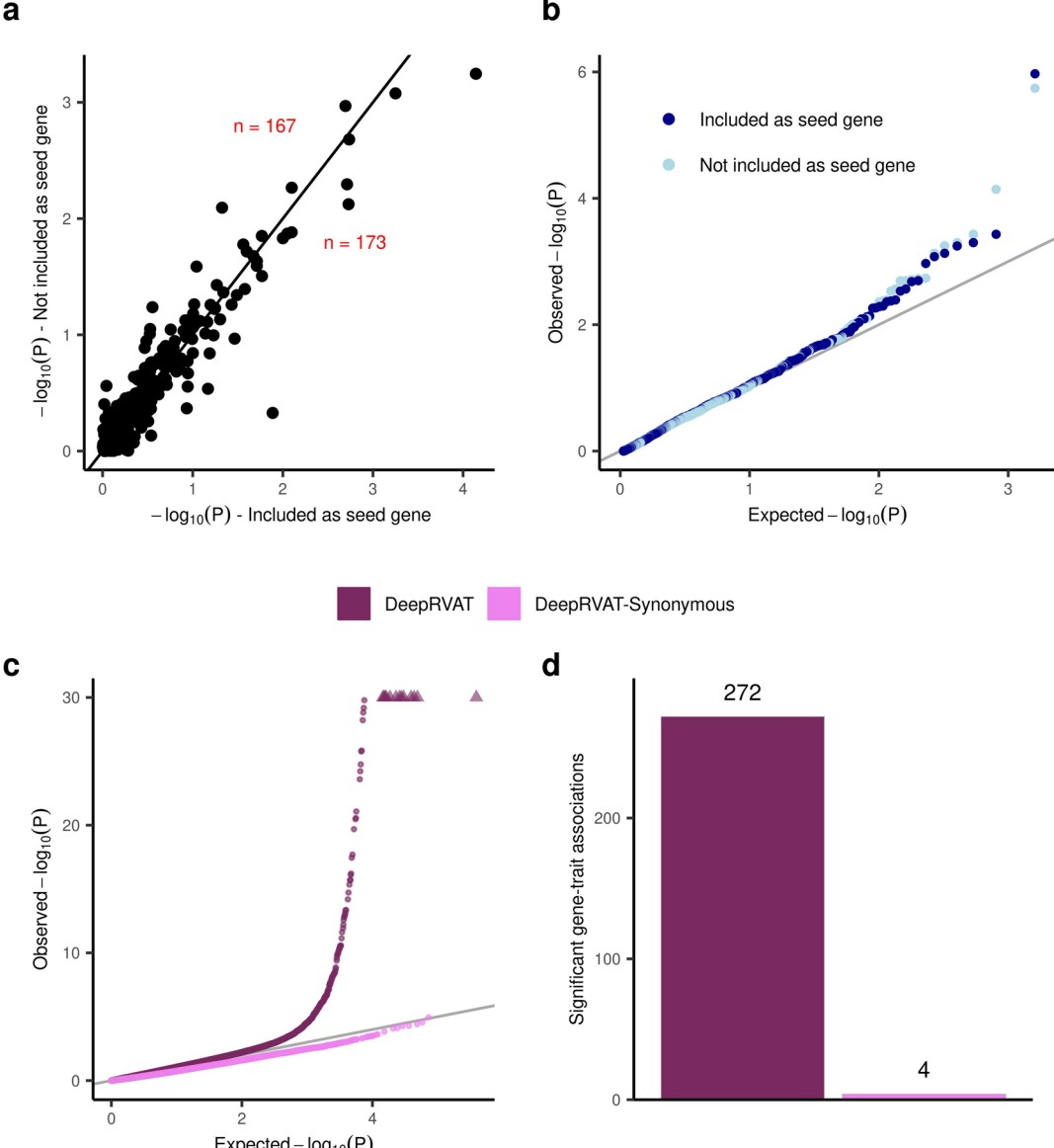

**Extended Data Fig. 2 | Robustness to inclusion of non-associated genes as seed genes and negative control. a,b,** Robustness of the DeepRVAT association test to the inclusion of non-associated genes as seed genes during training. DeepRVAT was trained 10 times using a seed gene set extended with 20% randomly sampled genes without evidence for association with the trait (P-value burden/SKAT combined > $10^{-4}$). Shown are P-values from a conventional DeepRVAT model versus P-values from DeepRVAT trained using the extended seed gene set, evaluated on the additionally included seed genes (pooled across restarts, 10 × 34 additionally included seed genes). **a,** Scatter plot of DeepRVAT P-values for non-associated seed genes, either considering the DeepRVAT model trained with (x-axis) or without (y-axis) the extended seed gene set. Red

numbers denote the number of tests above and below the diagonal. **b,** Q–Q plot of observed vs. expected (under the null hypothesis) DeepRVAT P-values for non-associated genes, either considering the DeepRVAT model trained with or without the extended seed gene set. **c,d,** Empirical assessment of DeepRVAT calibration using synonymous variants as negative controls. Results for DeepRVAT-synonymous were obtained from the application of the pre-trained DeepRVAT gene impairment module trained on all variant types applied to the same 21 training traits, however restricting to synonymous variants during test time. **c,** Q–Q plot of observed vs. expected (under the null hypothesis) unadjusted P-values across all 21 traits. **d,** Cumulative number of gene–trait associations (FWER < 5%) across all 21 traits.

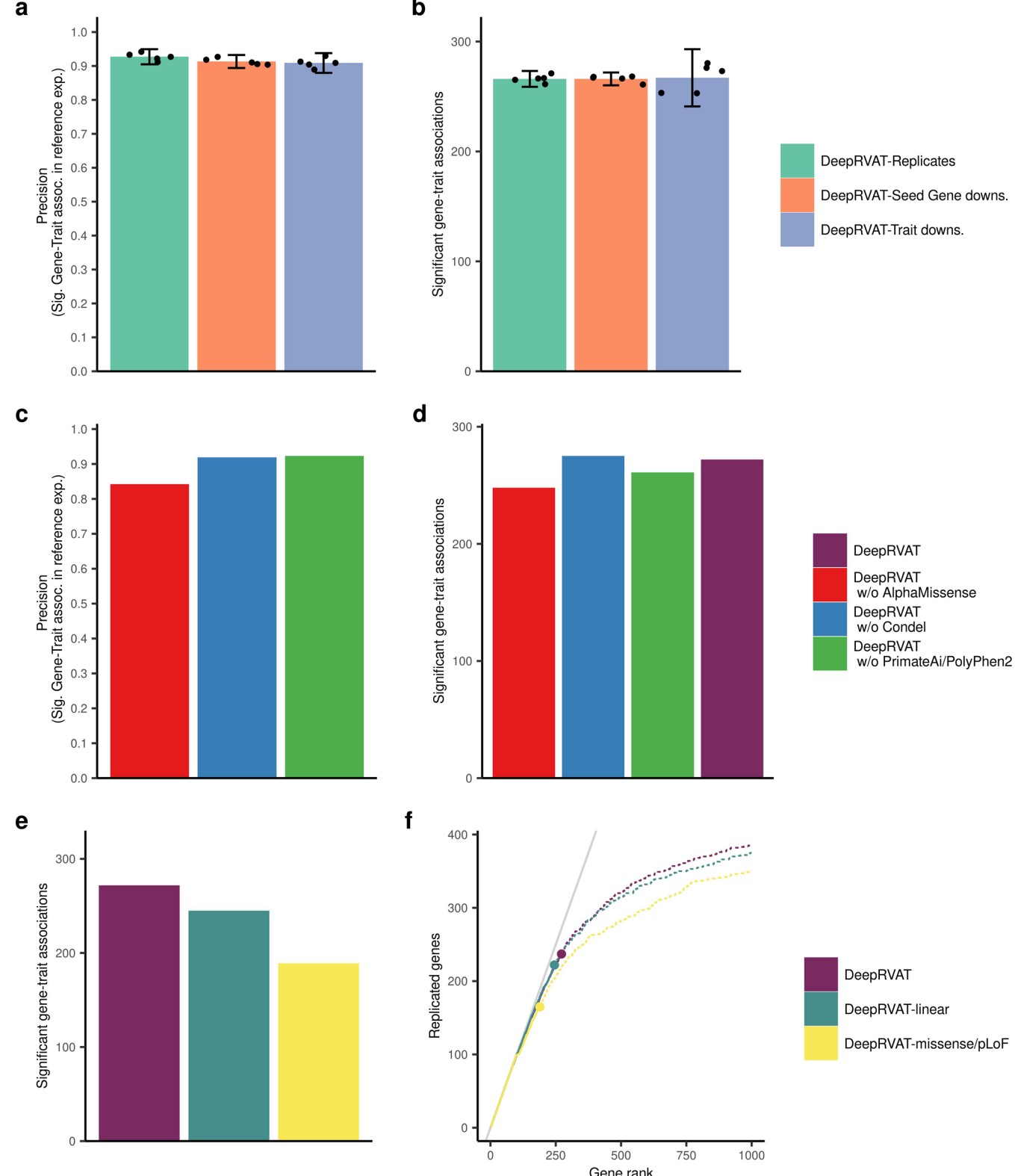

**Extended Data Fig. 3 | See next page for caption.**

**Extended Data Fig. 3 | Sensitivity of DeepRVAT to changes in training data and model architecture. a,b**, To assess sensitivity to differences in the training data, we modified the training dataset, including downsampling of seed genes (DeepRVAT-seed gene downs.; 10% randomly removed for each trait) and downsampling of traits (DeepRVAT-trait downs.; 20%, that is, 4 traits randomly removed per run). As a reference point, we considered the intrinsic variability of DeepRVAT by training the model 5 times (DeepRVAT-replicates). All models were applied to the 21 traits as considered in DeepRVAT training; results from the full model as shown in Fig. 2a were considered as a reference result set. Error bars denote 95% confidence intervals of the mean across 5 replicates. **a**, Precision, defined as the proportion of discoveries shared with the reference DeepRVAT results. **b**, Cumulative number of gene–trait associations (FWER < 5%) across all 21 traits. **c,d**, Analogous analyses as in **a** and **b**, but excluding certain features highly correlated with some others (AlphaMissense, Condel, and PrimateAI and Polyphen2; see Supplementary Fig. 7). Shown are results from a single DeepRVAT run. **e,f**, Comparison of the full DeepRVAT model to alternative configurations, considering a minimalistic set of annotations (MAF, pLOF and missense status: DeepRVAT-missense/pLOF), or a reduced model architecture (linear gene impairment module: DeepRVAT-linear). **e**, Cumulative number of gene–trait associations discovered by alternative DeepRVAT configurations across 21 traits (FWER < 5%). **f**, Replication of the cumulative discoveries as in **a**, in larger cohorts (cohorts and plot defined as in Fig. 3b) across 21 traits.

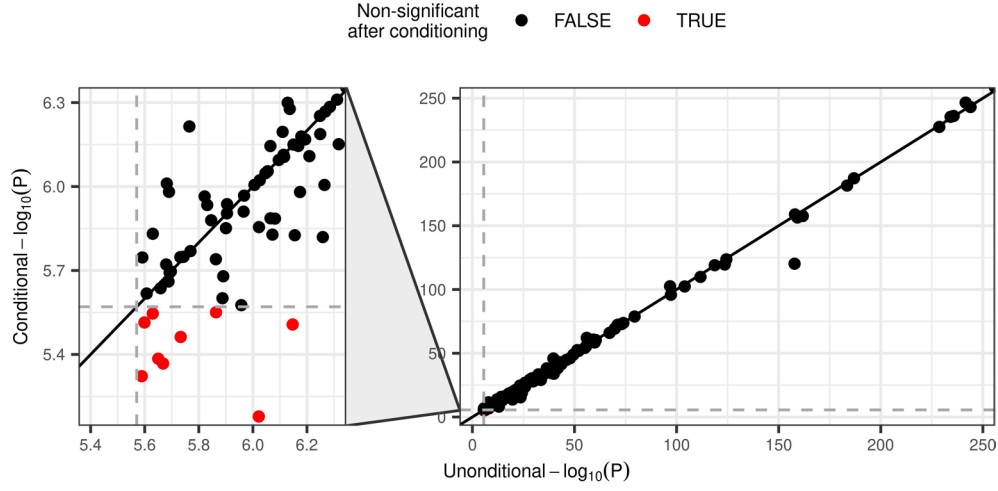

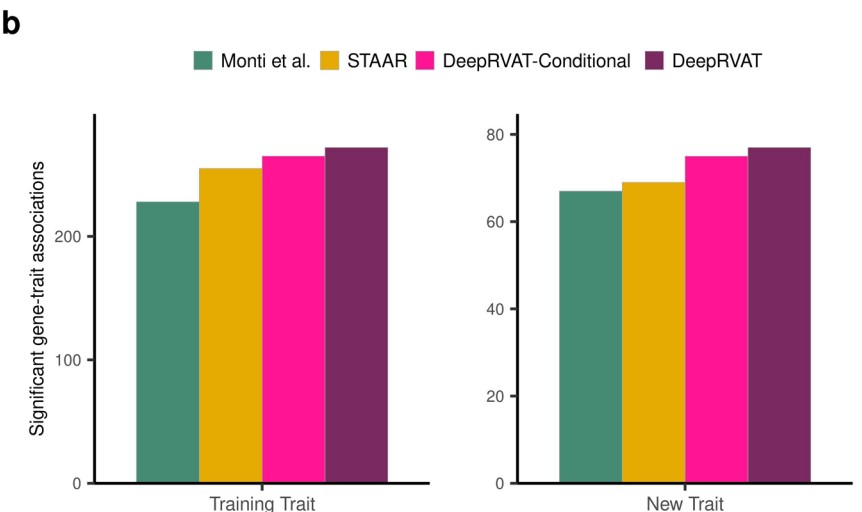

**Extended Data Fig. 4 | DeepRVAT association analysis conditioned on common variant effects.** Assessment of the effect of conditioning on common variant effects in the DeepRVAT association test. Common variants independently associated with a trait were identified from Pan-UK Biobank[59] GWAS summary statistics, derived using LD-based clumping using PLINK v1.9 with default parameters, considering associations with a $P$-value $< 10^{-7}$ and MAF $> 1\%$. Independently associated variants within ±500 kb of the gene boundaries were included as covariates during association testing. **a**, Scatter plot of unadjusted $p$-values, comparing DeepRVAT association tests with and without conditioning on common variants, considering significant gene–trait associations as in Fig. 2a,f (FWER < 5%). **b**, Cumulative number of gene–trait associations discovered by alternative methods (FWER < 5%), either considering the set of 21 traits used during DeepRVAT training (left) or 13 additional traits not used during training (right; Fig. 2a,f).

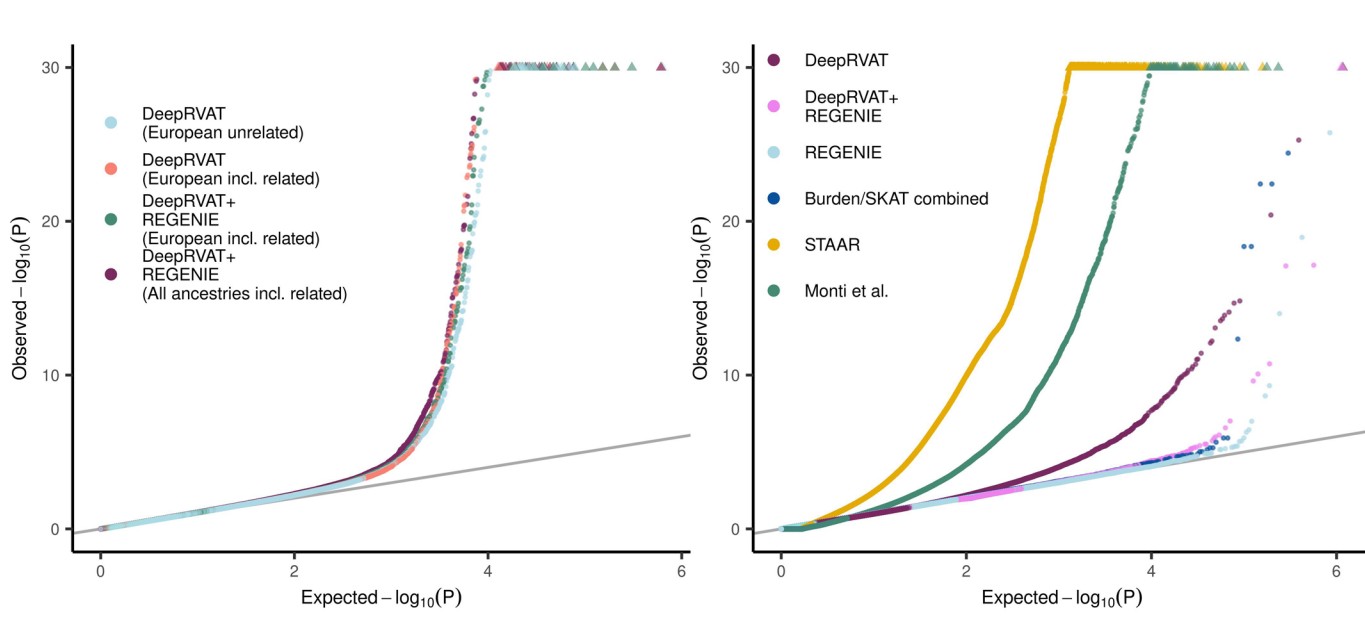

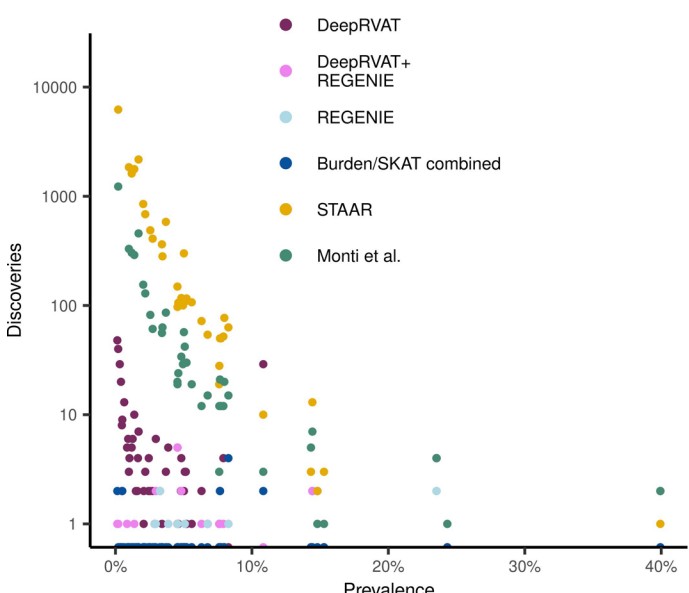

**Extended Data Fig. 5 | Application of DeepRVAT with REGENIE to the full UK Biobank dataset and imbalanced case–control settings. a**, Q–Q plots of expected vs. observed unadjusted association testing *p*-values across 34 quantitative traits, considering either the standard DeepRVAT test (DeepRVAT) or the combination DeepRVAT and REGENIE (DeepRVAT + REGINIE), using DeepRVAT scores as dosages. Shown are results obtained on different sub-cohorts of the UKBB 200k WES release: unrelated individuals of European ancestry (European unrelated), all individuals of European ancestry (European incl. related) and all 200k individuals (all ancestries incl. related). **b,c**, Application

of alternative methods to 63 binary traits, considering unrelated individuals of European ancestry from the 200k WES release. Results analogous to Fig. 4b,c, however additionally considering the standard DeepRVAT test without REGINIE (DeepRVAT) and burden/SKAT combined. **b**, Q–Q plot of observed versus expected unadjusted association testing *p*-values across all 63 binary traits. **c**, Scatter plot of the number of gene–trait associations versus the trait prevalence for each of 63 binary traits. **b**, Q–Q plots of raw association testing *p*-values across all binary traits. **c**, Number of gene–trait associations for each trait, plotted as a function of trait prevalence.

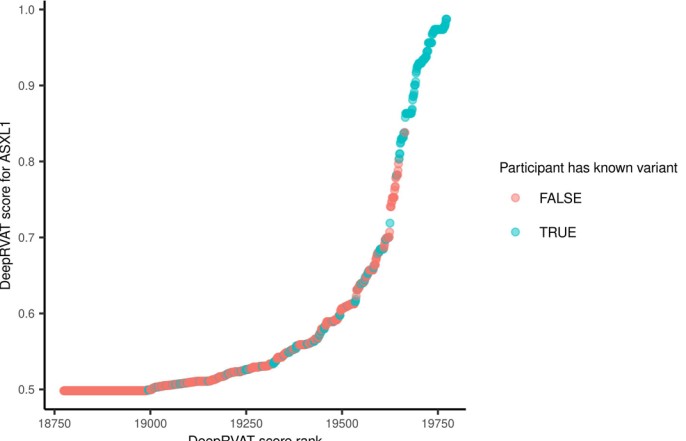

**Extended Data Fig. 6 | Largest DeepRVAT scores for *ASXL1* in UK Biobank participants diagnosed with heart failure.** Shown is, for 19,773 UK Biobank participants from the full 470k WES diagnosed with heart failure, the DeepRVAT gene impairment score for the *ASXL1* gene as a function of their rank. Each point corresponds to a participant; the figure is restricted to participants with the top 1,000 scores. Participants that carry a known mutation in *ASXL1* linked to clonal hematopoiesis (CH)[66,67], or lymphoid and myeloid cancer[68–70] are indicated in light blue. While the disease status of the top-scoring individuals can be explained by these known variants, many moderate-to-high scores cannot be explained by known variants.

# Reporting Summary

## Statistics

For all statistical analyses, confirm that the following items are present in the figure legend, table legend, main text, or Methods section.

| n/a | Confirmed | |
|---|---|---|
| ☐ | ☒ | The exact sample size (*n*) for each experimental group/condition, given as a discrete number and unit of measurement |
| ☐ | ☒ | A statement on whether measurements were taken from distinct samples or whether the same sample was measured repeatedly |
| ☐ | ☒ | The statistical test(s) used AND whether they are one- or two-sided *Only common tests should be described solely by name; describe more complex techniques in the Methods section.* |
| ☐ | ☒ | A description of all covariates tested |
| ☐ | ☒ | A description of any assumptions or corrections, such as tests of normality and adjustment for multiple comparisons |
| ☐ | ☒ | A full description of the statistical parameters including central tendency (e.g. means) or other basic estimates (e.g. regression coefficient) AND variation (e.g. standard deviation) or associated estimates of uncertainty (e.g. confidence intervals) |
| ☐ | ☒ | For null hypothesis testing, the test statistic (e.g. $F$, $t$, $r$) with confidence intervals, effect sizes, degrees of freedom and $P$ value noted *Give P values as exact values whenever suitable.* |
| ☒ | ☐ | For Bayesian analysis, information on the choice of priors and Markov chain Monte Carlo settings |
| ☒ | ☐ | For hierarchical and complex designs, identification of the appropriate level for tests and full reporting of outcomes |
| ☐ | ☒ | Estimates of effect sizes (e.g. Cohen's *d*, Pearson's *r*), indicating how they were calculated |

*Our web collection on statistics for biologists contains articles on many of the points above.*

## Software and code

Policy information about availability of computer code

| | |
|---|---|
| Data collection | No data collection was performed. No software was used for data collection. |
| Data analysis | DeepRVAT(v1.0) was used in both simulation and real data analysis and is implemented as an open-source python package available at https://github.com/PMBio/deeprvat/. Pre-trained DeepRVAT models are available on Zenodo: https://doi.org/10.5281/zenodo.12772611. The code for downstream analyses is available at  https://github.com/PMBio/deeprvat-analysis/. For association testing, the SEAK package (0.4.3) was used https://seak.readthedocs.io/en/latest/. Data analysis was performed using python (3.8) and R (4.2). All modules used for DeepRVAT training and data postprocessing are listed in https://github.com/PMBio/deeprvat/blob/main/deeprvat_env.yaml. Modules required for analyses in R are provided here https://github.com/PMBio/deeprvat-analysis/blob/main/r-env.yaml. UK Biobank data preprocessing was done using our custom preprocessing pipeline available at https://github.com/PMBio/deeprvat/blob/main/deeprvat/preprocessing/README.md using bcftools (1.10.2) and samtools (1.9). The bgen package  (v1.6.1) was used to convert the (samples  genes) matrix of DeepRVAT scores to a BGEN file of pseudovariants. Association testing using REGENIE was done using the REGENIE package (v3.4) https://github.com/rgcgithub/regenie/ Independently associated variants were identified from GWAS summary statistics through LD-based clumping using PLINK  (v1.9). DeepSEA predictions were obtained using Kipoi-veff2 (https://github.com/kipoi/kipoi-veff2). DeepRIPE predictions were obtained using the code provided here https://github.com/ohlerlab/DeepRiPe. All remaining variant annotations (excluding AbSplice, SpliceAI and AlphaMissense) were obtained using VEP v109. |

For manuscripts utilizing custom algorithms or software that are central to the research but not yet described in published literature, software must be made available to editors and reviewers. We strongly encourage code deposition in a community repository (e.g. GitHub). See the Nature Portfolio guidelines for submitting code & software for further information.

## Data

All manuscripts must include a <u>data availability statement</u>. This statement should provide the following information, where applicable:

- Accession codes, unique identifiers, or web links for publicly available datasets
- A description of any restrictions on data availability
- For clinical datasets or third party data, please ensure that the statement adheres to our <u>policy</u>

AlphaMissense scores were obtained from https://www.google.com/url?q=https://storage.googleapis.com/dm_alphamissense/
AlphaMissense_hg38.tsv.gz&sa=D&source=editors&ust=1714037736792761&usg=AOvVaw34CM435oT9SM5ziM6SQn2-.
AbSplice and SpliceAI scores were obtained from Zenodo (https://zenodo.org/record/6631476).
PRS Scores were obtained from https://www.pgscatalog.org/ using PRS ids provided in Supplementary Table 3.
GWAS summary statistics were obtained from the Pan-UK Biobank resource (https://pan.ukbb.broadinstitute.org , Karczewski et al., medRxiv 2024).
The UK Biobank analyses were conducted using the UK Biobank resource (Project IDs 25214, 44108, and 81358).
Replication data was retrieved from genebass gs://ukbb-exome-public/500k/results/results.mt and and the study by Backman et al., 2021 (https://doi.org/10.1038/
s41586-021-04103-z) (https://static-content.springer.com/esm/art%3A10.1038%2Fs41586-021-04103-z/MediaObjects/41586_2021_4103_MOESM5_ESM.xlsx
SD2) and is also provided in Supplementary Table 7.
GENCODE release 38 can be downloaded from https://www.gencodegenes.org/human/release_38.html
The association testing results from DeepRVAT+REGENIE on the 500k UK Biobank dataset, covering all genes and traits for all ancestries and Caucasians only, are
available on Zenodo https://doi.org/10.5281/zenodo.12736824.

# Research involving human participants, their data, or biological material

| | |
|---|---|
| Reporting on sex and gender | The genetic sex of participants was accessed through UK-Biobank Data-Field 22001, and included as a covariate in the statistical analyses. The genetic sex was determined by a genotyping analysis as described here: https://biobank.ndph.ox.ac.uk/showcase/field.cgi?id=22001<br><br>Of the 469,835 individuals considered in this study, 254,489 were estimated to be female and 214,893 estimated to be male by the genotyping analysis. We note that the estimated genetic sex can differ from the self-reported sex from UK Biobank Data Field 31. |
| Reporting on race, ethnicity, or other socially relevant groupings | To minimize confounding due to population structure, we restricted to 161,822 unrelated individuals of Caucasian genetic ethnicity as determined by an analysis of genetic principal components for our benchmarking analyses. For biological discovery, we used individuals of all ancestry from the UK Biobank. |
| Population characteristics | The mean age at recruitment of the 469,835 participants included in the statistical analyses in this study was 56.54, standard deviation 8.10, and ranged from 37 to 73 years. The average BMI in this sample was 27.47 (standard deviation 4.77). |
| Recruitment | The UK Biobank recruited approximately 500,000 individuals from 2006 to 2010 with a target age of 40-69 by mailers to people in the UK medical system. Informed consent was obtained by the UK Biobank for all participants. |
| Ethics oversight | The scientific protocol of the UK Biobank is approved by appropriate external ethics committees in accordance with guidance from relevant bodies. Instead of requiring each applicant to obtain separate ethics approval, UK Biobank has sought generic Research Tissue Bank (TB) approval, which covers the large majority of research using the resource.<br>The original approval for the UK Biobank was granted in 2011 by the National Research Ethics Service (NRES) Committee North West - Haydock. The approval was renewed in 2016 and 2021 by the Health Research Authority, North West - Haydock Research Ethics Committee.<br>For additional information, see https://www.ukbiobank.ac.uk/learn-more-about-uk-biobank/about-us/ethics.<br>This research has been conducted using the UK Biobank Resource under Application Numbers 25214, 44108, and 81358. UKBB participants received no compensation. |

Note that full information on the approval of the study protocol must also be provided in the manuscript.

# Field-specific reporting

Please select the one below that is the best fit for your research. If you are not sure, read the appropriate sections before making your selection.

☒ Life sciences          ☐ Behavioural & social sciences          ☐ Ecological, evolutionary & environmental sciences

For a reference copy of the document with all sections, see <u>nature.com/documents/nr-reporting-summary-flat.pdf</u>

# Life sciences study design

All studies must disclose on these points even when the disclosure is negative.

| | |
|---|---|
| Sample size | Sample sizes were determined from the data. We did not pre-define sample sizes based on power estimates. The sample size for each |

| | |
|---|---|
| Sample size | phenotype in this study was determined by the number of individuals which had both complete (i.e., non-missing) phenotype and covariate data. Because of varying levels of missingness for the different phenotypes, the sample size ranged from 406478 to 468386 samples for quantitative traits. Binary traits were extracted using the definitions by Jurgens et al. 2022, Supp. Table 1 (https://doi.org/10.1038/s41588-021-01011-w) and considered as cases if they had a matching code according to the trait definitions. All remaining samples were considered as controls (if not matching an 'exclude' code). |
| Data exclusions | We removed individuals who had withdrawn consent. To minimize confounding due to population structure and population structure, we restricted to 161,822 unrelated individuals of Caucasian genetic ethnicity as determined by an analysis of genetic principal components for our benchmarking analyses. For biological discovery, we used individuals of all ancestry from the UK Biobank.<br>For binary traits, samples matching an 'exclude' code as defined by the trait definitions were excluded for the respective trait. |
| Replication | Significant associations for quantitative traits were compared to two studies on a larger cohort from the UK Biobank that employed conventional RVAT strategies (Backman et al., 2021 and Karczewski et al., 2022/Genebass, see 'Data'). For binary traits, we also compared to Jurgens et al., 2022.  The replication rate of identified gene-trait associations is shown in Fig. 2c,g for quantitative traits. For binary traits, replication assessment was complicated by variable phenotype definitions across studies and is provided in Supp. Table 9. |
| Randomization | We did not allocate samples into experimental groups. No randomization was performed. |
| Blinding | No groups were allocated. |

# Reporting for specific materials, systems and methods

We require information from authors about some types of materials, experimental systems and methods used in many studies. Here, indicate whether each material, system or method listed is relevant to your study. If you are not sure if a list item applies to your research, read the appropriate section before selecting a response.

## Materials & experimental systems

| n/a | Involved in the study |
|---|---|
| ☒ | Antibodies |
| ☒ | Eukaryotic cell lines |
| ☒ | Palaeontology and archaeology |
| ☒ | Animals and other organisms |
| ☒ | Clinical data |
| ☒ | Dual use research of concern |
| ☒ | Plants |

## Methods

| n/a | Involved in the study |
|---|---|
| ☒ | ChIP-seq |
| ☒ | Flow cytometry |
| ☒ | MRI-based neuroimaging |

## Plants

| | |
|---|---|
| Seed stocks | *Report on the source of all seed stocks or other plant material used. If applicable, state the seed stock centre and catalogue number. If plant specimens were collected from the field, describe the collection location, date and sampling procedures.* |
| Novel plant genotypes | *Describe the methods by which all novel plant genotypes were produced. This includes those generated by transgenic approaches, gene editing, chemical/radiation-based mutagenesis and hybridization. For transgenic lines, describe the transformation method, the number of independent lines analyzed and the generation upon which experiments were performed. For gene-edited lines, describe the editor used, the endogenous sequence targeted for editing, the targeting guide RNA sequence (if applicable) and how the editor was applied.* |
| Authentication | *Describe any authentication procedures for each seed stock used or novel genotype generated. Describe any experiments used to assess the effect of a mutation and, where applicable, how potential secondary effects (e.g. second site T-DNA insertions, mosiacism, off-target gene editing) were examined.* |

