## [Peer Review File · Nature Genetics]

Peer Review Information

Manuscript Title: Integration of variant annotations using deep set networks boosts rare variant association testing

Corresponding author name(s): Dr Brian Clarke, Professor Oliver Stegle, Professor Julien Gagneur

Reviewer Comments & Decisions:

Decision Letter, initial version:

6th Nov 2023

Dear Oliver,

Your Technical Report, "Integration of variant annotations using deep set networks boosts rare variant association genetics" has now been seen by 3 referees. You will see from their comments copied below that while they find your work of considerable potential interest, they have raised quite substantial concerns that must be addressed. In light of these comments, we cannot accept the manuscript for publication, but would be very interested in considering a revised version that addresses these serious concerns.

In brief, there is support voiced for the application of DL to RVAT that DeepRVAT presents, but it is also clear that the broad utility of your tool does not yet reach the standard required for publication at the journal.

Reviewer #1 thinks this is a well-presented study but highlights several critical technical concerns, e.g. inability to control for sample relatedness, and thinks the overall biological novelty is also lacking. They do, however, sound open to seeing a revision. Referees #2 and #3, conversely, are more positive, but make a number of thoughtful comments most strikingly, the shared comment regarding the use of seed genes for model training.

In our reading of these reviews, we think there is a path to publication, but there are important and fundamental technical issues that require complete clarification. Most notable of these is the concern about the seed genes and how this affects DeepRVAT and interpreting results; we also think Reviewer #1's comments on how DeepRVAT will actually be used in biobank-scale data (i.e. accounting for sample relatedness), and a better demonstration of the ability of DeepRVAT to enable novel biological discovery, will need to be fully addressed for them to support publication.

We hope you will find the referees' comments useful as you decide how to proceed. If you wish to submit a substantially revised manuscript, please bear in mind that we will be reluctant to approach

the referees again in the absence of major revisions.

To guide the scope of the revisions, the editors discuss the referee reports in detail within the team, including with the chief editor, with a view to identifying key priorities that should be addressed in revision and sometimes overruling referee requests that are deemed beyond the scope of the current study. We hope that you will find the prioritised set of referee points to be useful when revising your study. Please do not hesitate to get in touch if you would like to discuss these issues further.

If you choose to revise your manuscript taking into account all reviewer and editor comments, please highlight all changes in the manuscript text file. At this stage we will need you to upload a copy of the manuscript in MS Word .docx or similar editable format.

*2) If you have not done so already please begin to revise your manuscript so that it conforms to our Technical Report format instructions, available here. Refer also to any guidelines provided in this letter.

Please be aware of our guidelines on digital image standards.

[redacted]

If you wish to submit a suitably revised manuscript we would hope to receive it within 6 months. If you cannot send it within this time, please let us know. We will be happy to consider your revision so long as nothing similar has been accepted for publication at Nature Genetics or published elsewhere. Should your manuscript be substantially delayed without notifying us in advance and your article is eventually published, the received date would be that of the revised, not the original, version.

Thank you for the opportunity to review your work.

Sincerely,

Michael Fletcher, PhD
Senior Editor, Nature Genetics

ORCID: 0000-0003-1589-7087

Referee expertise: statistical and human genetics, including rare variant association testing.

Reviewers' Comments:

Reviewer #1:

Remarks to the Author:

In this paper, the authors describe DeepRVAT, a method for rare variant association test (RVAT) by integrating functional annotations. DeepRVAT builds upon deep neural networks to estimate a gene impairment score, which could then be used in RVAT and phenotype prediction. The authors used UK Biobank WES data to illustrate the proposed method in rare variant association detection and phenotype prediction. The manuscript is concise, and most parts of the paper are well-written. However, it lacks clarity on several points, and some of the main analysis results do not sufficiently support the conclusions drawn by the authors.

I do have some major concerns that I would like the authors to address.

(1) The proposed DeepRVAT method could not control sample relatedness, which limited the ability to analyze whole-genome/exome sequencing data of UK Biobank and others, such as TOPMed1 and All of Us2. The existing RVATs either use the mixed effect model (for example, STAAR3) or a covariate of polygenic effect (for example, REGENIE4) to control for population stratifications and relatedness. In addition, the authors use the European-ancestry only UK Biobank data to illustrate the proposed method. This reviewer recommends the authors to extend the proposed method to account for population stratifications and sample relatedness as part of the current work.

(2) The authors only use the total number of significant associations to demonstrate the advantage of the proposed method. These claims would not be necessarily helpful. One would want to see how many putatively new genes or masks of rare variants can be detected using each method. To identify putatively new rare variant associations, one possible way is to perform conditional analysis of each significant association by adjusting for known variants nearby5.

(3) This reviewer did not observe any inflation or deflation of STAAR for quantitative from the Q-Q plot in Fig.3d and Supp. Fig. 3.3a. It is not meaningful to use the genomic inflation factor lambda to evaluate the calibration of STAAR. STAAR uses the ACAT method to integrate the p-values in the

weighting scheme to incorporate multiple functional annotations, and the ACAT method gives accurate p-values for the tail probability⁶.

(4) Following (3), as Fig. 3a showed that STAAR detected slightly more significant associations than DeepRVAT for 21 training traits. This reviewer recommends that the author perform more analysis to provide additional evidence or only claim that the proposed method has comparable performance as STAAR for quantitative traits.

(5) The results of significant gene-trait associations (Supp. Table 6) are misleading. Many of the “new DeepRVAT discovery” findings are known and previously reported. Specifically, nearly all claimed “new DeepRVAT discovery” findings of lipid traits are known lipid associated genes. For example, the p-value of the association between RVs in NPC1L1 and DeepRVAT is 5.59E-07. However, the STAAR p-value of disruptive missense RVs or missense RVs or plof and disruptive missense RVs in NPC1L1 are all significantly associated with LDL using UK Biobank 200k WES data, with p-values 1.78E-07, 5.20E-09, and 3.97E-09 (see Supplementary data 14 in Selvaraj et al⁷). I am a bit confused why the authors claimed that other methods could not detect these genes.

Other comments

(6) Following (5), The results of significant gene-trait associations (Supp. Table 6) are not clear. These results could not be claimed as new discoveries, as many of the results are not well calibrated for the necessary covariates. For example, in addition to age, sex and ancestry PCs, lipid traits need to account for the statin usage^{7,8}. This comment is also related to the main concern 2a. The total number of significant findings in the manuscript is not biologically meaningful.

Reference

1. Taliun, D. et al. Sequencing of 53,831 diverse genomes from the NHLBI TOPMed Program. *Nature* 590, 290-299 (2021).
2. Investigators, A.o.U.R.P. The “All of Us” research program. *New England Journal of Medicine* 381, 668-676 (2019).
3. Li, X. et al. Dynamic incorporation of multiple in silico functional annotations empowers rare variant association analysis of large whole-genome sequencing studies at scale. *Nature genetics* 52, 969-983 (2020).
4. Mbatchou, J. et al. Computationally efficient whole-genome regression for quantitative and binary traits. *Nature Genetics* 53, 1097-1103 (2021).
5. Backman, J.D. et al. Exome sequencing and analysis of 454,787 UK Biobank participants. *Nature* 599, 628-634 (2021).
6. Liu, Y. et al. ACAT: a fast and powerful p value combination method for rare-variant analysis in sequencing studies. *The American Journal of Human Genetics* 104, 410-421 (2019).
7. Selvaraj, M.S. et al. Whole genome sequence analysis of blood lipid levels in >66,000 individuals. *Nature Communications* 13, 5995 (2022).
8. Hindy, G. et al. Rare coding variants in 35 genes associate with circulating lipid levels—A multi-ancestry analysis of 170,000 exomes. *The American Journal of Human Genetics* 109, 81-96 (2022).

Reviewer #2:

Remarks to the Author:

Clarke, Holtkamp, et al. propose DeepRVAT, a data-driven deep learning neural network for rare variant analyses built from a variety of variant annotations. DeepRVAT essentially computes a gene impairment score by integrating a phenotype module and a dynamic set of variant annotations, which

accounts for the nonlinear effects of rare variants, imbalanced case-control ratio of binary traits and applies to multiple phenotypes. With the gene impairment score, DeepRVAT not only improves gene discoveries through their adaptive rare variant association tests but also aids in phenotype prediction that incorporates rare variant effects. DeepRVAT appears to be more computationally efficient enough to be applied to biobank level data, which would facilitate the downstream analysis of rare variants from a broader perspective and at a larger scale.

The manuscript is well-organized and clearly written. The data and results provided in the text adequately demonstrate the accuracy and efficiency of DeepRVAT compared to the previous RVAS methods. This new method would serve as a valuable complement to the traditional burden tests and approaches for PRS-based phenotype prediction, in that it effectively incorporates more information from variant annotations and can be highly generalizable in terms of both variant and phenotype information, which will help advance the understanding of rare variants and complex disease. Here are some comments that I hope would improve the manuscript:

Major comments:

I applaud the authors for building a simulation framework that attempts to balance many parameters in simulating true signals among noise. However, when digging into Supp. Fig. 2.1, it seems that LoF-like variants (e.g. frameshifts) seem quite a bit underspecified and have fewer proportion causal than missense. Are the effect sizes modeled more strongly among pLOF variants as one would expect? Put another way, when running the burden tests on these simulated variants, are results comparable to those in UK Biobank which show increased signal among pLOF variants above missense? This is somewhat observed in Fig. 2a where the significance of pLOF is lower than missense and in 2b where the missense burden is higher powered than the pLOF, which is inconsistent with the Backman and Genebass analyses where most signals stem from pLOF. That they get relatively higher importance values in the real data (Supp. Fig. 3.5) is encouraging, but the simulation framework seems a bit misspecified in this regard.

Similarly, it may be advantageous to use synonymous variants as a sort of negative class. Of course, some synonymous variants may be associated due to LD, but this might give some level of baseline expectation.

The authors mentioned in the Discussion section that it is challenging to choose the most appropriate set of annotations. Would it be possible for the authors to provide some general recommendations for phenotype and variant annotation selection, as well as parameter configurations (number of repeats, number of seed genes, etc.) to users who would want to customize their own model?

How are the association results for the seed genes and target phenotypes used in training the gene impairment scores obtained using DeepRVAT, if they cannot be included in the association testing module?

Given that the seed genes are selected from those with associations with phenotypes of interest, are the seed genes supposed to be associated with all selected phenotypes or any phenotype from the target set? Also, will there be a default set of seed genes provided for users?

The description of the model is a bit confusing at times. Figure 1 suggests that only annotations are used in the gene impairment model, but later "We used 21 quantitative traits [...] to train DeepRVAT". Can you clarify this, and if indeed phenotypes are used, elaborate more on how this model can be generalized to any other phenotypes with no restrictions, more specifically, what makes the gene impairment score both gene- and trait-agnostic?

How will DeepRVAT generalize to cohorts with multiple ancestry groups?

The authors note that "the overall contribution of rare variants was modest" which is very much in line with expectations. Have they considered this separately for each phenotype? Specifically, there are

new methods that estimate heritability contribution from rare variants (Weiner, Nadig, et al, 2023) - are the contributions from the rare variants in this model correlated with burden heritability results? Supp. Fig. 3.1: I'm a bit surprised that MAF is not correlated with other features. Is this an artifact of Pearson correlation being not ideal for a linearly compressed metric like MAF? Would Spearman correlation (or $\log_{10}(\text{AF})$) be better here?

Similarly, there seems to be a group of highly correlated features included in model training. Can the authors comment on how such inclusion will impact the performance of the model?

In Supp. Fig. 3.7 and Methods 3.2, when repeating for more than one time, what are the parameters passing into the next repeat? If there are not any elements inherited, how are the uncorrected p-values from each repeat being processed and aggregated?

Why is DeepRVAT specific to rare variants? Are there any assumptions that prevent applying it to common variants?

Minor comments:

The authors claimed that DeepRVAT presents well-calibrated results for imbalanced binary traits. It would be helpful to report the number of cases and controls for the binary traits analyzed in this study and compare the performance of different models across different case/control ratios to validate the statement.

The authors should clarify the default proportion of causal variants for simulation in Figure 2 in the main text. Similarly, in Figure 2a, it is hard to see the causal variants among the non-causal ones. Comparing the mean chi-squared (or similar) for causal vs non-causal may make this point more strongly.

It took me a while to understand what the rows in Figure 2b represented - perhaps a simplified label (high, medium, no influence of ultra-rare variants, or something like that) would help orient.

The authors should add or cite results related to this statement "... including in settings for which additional non-causal variants were incorporated in the MAF cutoff..." in the "Model validation using simulated data" section.

Were the traits used for training chosen in some non-random way? They seem to have on average more associations (Fig. 3a) than those for evaluation (3e). I would understand if there's some light overfitting happening, but the Burden/SKAT combined also shows lower numbers, which doesn't have nearly as much influence of the traits used. Additionally, can the authors report replication results for the evaluation traits used in Fig 3d-3f (with a similar display as in Fig 3b)?

The authors should label the panels of Figure 4b and 4c.

What are the dotted lines in Figure 4c?

Supp. Table S2, instead of Supp. Table S4 should be cited in the caption of Supp. Fig. 2.1.

Supp. Fig. 2.1 is missing captions for panels a and b, specifically for panel b, a short description of the metric in each panel might be helpful in explaining the different trends between causal and non-causal variants.

Supp. Fig. 2.1e is missing a color legend for the dark blue boxes.

In Supp. Fig 3.6, the authors should label the panels and avoid using "rows" and "columns" in the caption if they do not refer to the rows and columns of the figure. Is each panel specific to a VEP annotation? These should be spelled out if so, and the caption clarified.

Can the authors provide an explicit list of the required input files/information for each module in the Github repo of DeepRVAT?

Reviewer #3:

Remarks to the Author:

Clarke, Holtkamp, et al. propose DeepRVAT, a data-driven deep learning neural network for rare

variant analyses built from a variety of variant annotations. DeepRVAT essentially computes a gene impairment score by integrating a phenotype module and a dynamic set of variant annotations, which accounts for the nonlinear effects of rare variants, imbalanced case-control ratio of binary traits and applies to multiple phenotypes. With the gene impairment score, DeepRVAT not only improves gene discoveries through their adaptive rare variant association tests but also aids in phenotype prediction that incorporates rare variant effects. DeepRVAT appears to be more computationally efficient enough to be applied to biobank level data, which would facilitate the downstream analysis of rare variants from a broader perspective and at a larger scale.

The manuscript is well-organized and clearly written. The data and results provided in the text adequately demonstrate the accuracy and efficiency of DeepRVAT compared to the previous RVAS methods. This new method would serve as a valuable complement to the traditional burden tests and approaches for PRS-based phenotype prediction, in that it effectively incorporates more information from variant annotations and can be highly generalizable in terms of both variant and phenotype information, which will help advance the understanding of rare variants and complex disease. Here are some comments that I hope would improve the manuscript:

Major comments:

1. I applaud the authors for building a simulation framework that attempts to balance many parameters in simulating true signals among noise. However, when digging into Supp. Fig. 2.1, it seems that LoF-like variants (e.g. frameshifts) seem quite a bit underspecified and have fewer proportion causal than missense. Are the effect sizes modeled more strongly among pLOF variants as one would expect? Put another way, when running the burden tests on these simulated variants, are results comparable to those in UK Biobank which show increased signal among pLOF variants above missense? This is somewhat observed in Fig. 2a where the significance of pLOF is lower than missense and in 2b where the missense burden is higher powered than the pLOF, which is inconsistent with the Backman and Genebass analyses where most signals stem from pLOF. That they get relatively higher importance values in the real data (Supp. Fig. 3.5) is encouraging, but the simulation framework seems a bit misspecified in this regard.
2. Similarly, it may be advantageous to use synonymous variants as a sort of negative class. Of course, some synonymous variants may be associated due to LD, but this might give some level of baseline expectation.
3. The authors mentioned in the Discussion section that it is challenging to choose the most appropriate set of annotations. Would it be possible for the authors to provide some general recommendations for phenotype and variant annotation selection, as well as parameter configurations (number of repeats, number of seed genes, etc.) to users who would want to customize their own model?
4. How are the association results for the seed genes and target phenotypes used in training the gene impairment scores obtained using DeepRVAT, if they cannot be included in the association testing module?
5. Given that the seed genes are selected from those with associations with phenotypes of interest, are the seed genes supposed to be associated with all selected phenotypes or any phenotype from the target set? Also, will there be a default set of seed genes provided for users?
6. The description of the model is a bit confusing at times. Figure 1 suggests that only annotations are used in the gene impairment model, but later "We used 21 quantitative traits [...] to train DeepRVAT". Can you clarify this, and if indeed phenotypes are used, elaborate more on how this model can be generalized to any other phenotypes with no restrictions, more specifically, what makes the gene impairment score both gene- and trait-agnostic?

7. How will DeepRVAT generalize to cohorts with multiple ancestry groups?
8. The authors note that “the overall contribution of rare variants was modest” which is very much in line with expectations. Have they considered this separately for each phenotype? Specifically, there are new methods that estimate heritability contribution from rare variants (Weiner, Nadig, et al, 2023) - are the contributions from the rare variants in this model correlated with burden heritability results?
9. Supp. Fig. 3.1: I’m a bit surprised that MAF is not correlated with other features. Is this an artifact of Pearson correlation being not ideal for a linearly compressed metric like MAF? Would Spearman correlation (or $\log_{10}(\text{AF})$) be better here?
10. Similarly, there seems to be a group of highly correlated features included in model training. Can the authors comment on how such inclusion will impact the performance of the model?
11. In Supp. Fig. 3.7 and Methods 3.2, when repeating for more than one time, what are the parameters passing into the next repeat? If there are not any elements inherited, how are the uncorrected p-values from each repeat being processed and aggregated?
12. Why is DeepRVAT specific to rare variants? Are there any assumptions that prevent applying it to common variants?

Minor comments:

1. The authors claimed that DeepRVAT presents well-calibrated results for imbalanced binary traits. It would be helpful to report the number of cases and controls for the binary traits analyzed in this study and compare the performance of different models across different case/control ratios to validate the statement.
2. The authors should clarify the default proportion of causal variants for simulation in Figure 2 in the main text. Similarly, in Figure 2a, it is hard to see the causal variants among the non-causal ones. Comparing the mean chi-squared (or similar) for causal vs non-causal may make this point more strongly.
3. It took me a while to understand what the rows in Figure 2b represented - perhaps a simplified label (high, medium, no influence of ultra-rare variants, or something like that) would help orient.
4. The authors should add or cite results related to this statement “... including in settings for which additional non-causal variants were incorporated in the MAF cutoff...” in the “Model validation using simulated data” section.
5. Were the traits used for training chosen in some non-random way? They seem to have on average more associations (Fig. 3a) than those for evaluation (3e). I would understand if there’s some light overfitting happening, but the Burden/SKAT combined also shows lower numbers, which doesn’t have nearly as much influence of the traits used. Additionally, can the authors report replication results for the evaluation traits used in Fig 3d-3f (with a similar display as in Fig 3b)?
6. The authors should label the panels of Figure 4b and 4c.
7. What are the dotted lines in Figure 4c?
8. Supp. Table S2, instead of Supp. Table S4 should be cited in the caption of Supp. Fig. 2.1.
9. Supp. Fig. 2.1 is missing captions for panels a and b, specifically for panel b, a short description of the metric in each panel might be helpful in explaining the different trends between causal and non-causal variants.
10. Supp. Fig. 2.1e is missing a color legend for the dark blue boxes.
11. In Supp. Fig 3.6, the authors should label the panels and avoid using “rows” and “columns” in the caption if they do not refer to the rows and columns of the figure. Is each panel specific to a VEP annotation? These should be spelled out if so, and the caption clarified.
12. Can the authors provide an explicit list of the required input files/information for each module in the Github repo of DeepRVAT?

Author Rebuttal to Initial comments

Response to referees

We thank the reviewers for their constructive and insightful comments. In addressing them, we have made major revisions to the DeepRVAT method, our analyses, and the manuscripts, in ways that have substantially improved the demonstration of the methodological added value and immediate impact of DeepRVAT.

The major changes are:

- **A refined scheme for training DeepRVAT's gene impairment module.** We now employ a rigorous cross validation scheme to avoid leakage of statistical evidence. This approach is robust with respect to the exact selection of the seed genes, and gives calibrated test statistics, regardless whether a gene has been included as a seed gene or not. A brief description of this scheme is provided in the first section of Results, with full details provided in Sections 3.1 and 4.5 of Methods.
- **Extended analysis and assessment of DeepRVAT on (imbalanced) binary traits.** We demonstrate how DeepRVAT can be applied to conduct well-powered and calibrated RVAT analyses even for imbalanced binary traits (new Figure 4b,c). This is achieved thanks to DeepRVAT's modular architecture, which separates trait-agnostic gene impairment scoring on the one hand from gene-trait association testing on the other hand. We show how gene-trait association testing can be performed with dedicated algorithms using DeepRVAT impairment scores— we employ REGENIE to this end. Full technical details on the integration of REGENIE are provided in Section 3.3 of Methods.
- **Expanded application of DeepRVAT for biological discovery.** We have applied the model to the full 470K UKBB WES release. Collaborating with Dr. Zhifen Chen, PI at the German Heart Centre, we report and discuss compelling novel gene discoveries for cardiovascular conditions and cataracts. The full results are described in the new final section of Results and in the new Table 1.

Major changes in the main text of the paper are marked by blue text.

Other notable changes include:

- Conditional analyses correcting for common variant effects (Results, section 2, paragraph 5).
- Rather than model "repeats," we now average DeepRVAT gene scores from multiple training runs (as in standard model ensembling) before association testing, leading to a single p -value per gene. We employ conservative Bonferroni adjustment to control for multiple testing across genes.
- Incorporation of AlphaMissense annotations into DeepRVAT and STAAR.

- Inclusion of additional covariates (age² and age*sex), and correction of lipid traits for statin usage (Methods, sections 4.2 and 4.4).
- The DeepRVAT gene impairment module is now trained only on unrelated individuals (Results, section 2, paragraph 1), and we provide additional analyses to assess the robustness of the model to cohorts with related individuals as well as population structure (Results, section 4, paragraphs 1 & 2).

For more details, please see the point-by-point response below. We thank you again for your contribution to improving our manuscript.

Reviewer #1:

Remarks to the Author:

In this paper, the authors describe DeepRVAT, a method for rare variant association test (RVAT) by integrating functional annotations. DeepRVAT builds upon deep neural networks to estimate a gene impairment score, which could then be used in RVAT and phenotype prediction. The authors used UK Biobank WES data to illustrate the proposed method in rare variant association detection and phenotype prediction. The manuscript is concise, and most parts of the paper are well-written. However, it lacks clarity on several points, and some of the main analysis results do not sufficiently support the conclusions drawn by the authors.

I do have some major concerns that I would like the authors to address.

R1.1 The proposed DeepRVAT method could not control sample relatedness, which limited the ability to analyze whole-genome/exome sequencing data of UK Biobank and others, such as TOPMed1 and All of Us2. The existing RVATs either use the mixed effect model (for example, STAAR3) or a covariate of polygenic effect (for example, REGENIE4) to control for population stratifications and relatedness. In addition, the authors use the European-ancestry only UK Biobank data to illustrate the proposed method. This reviewer recommends the authors to extend the proposed method to account for population stratifications and sample relatedness as part of the current work.

We thank the reviewer for bringing up this very important point. We agree that accounting for population stratification and relatedness, as well as performing multi-ancestry analysis, should be addressed in DeepRVAT. We note in DeepRVAT there are two separate aspects, (i) model training and (ii) downstream applications, in particular association testing.

First, regarding the gene impairment module training, we have now restricted the training dataset to unrelated individuals up to the 3rd degree, removing about 5,000 from the 200,000 individuals, thus ensuring that relatedness does not affect the gene impairment module. For consistency, and to ensure that our benchmark is not confounded to varying extent to which different methods control for

relatedness, we use this filtered dataset for all benchmarks we consider (updated main text Fig. 2&3 and associated supplementary figures).

Second, regarding the robustness of association testing to related samples, we have created a workflow that integrates DeepRVAT gene impairment scores into REGENIE for gene-trait association testing (Results Section 4). Technically, this is achieved by creating a single pseudovariant per gene, with DeepRVAT scores as the dosage. We have included corresponding comparisons, demonstrating:

- Robust control for related individuals and multi-ancestry, resulting in improved replication in held-out data (Fig. 4a; Results, Section 4, paragraph 2) and statistical calibration (Supp. Fig. 4.1a).
- The combination of DeepRVAT with REGENIE entails a further substantial benefit for binary traits, where REGENIE is known to add value¹. We find that the combination of a data-driven gene impairment score provided by DeepRVAT with REGENIE allows to effectively address the pathological behavior of alternative association tests in the regime of low prevalence (Fig. 4b,c; Results, Section 4, paragraph 2, Supp. Fig. 4.1b).

Furthermore, we have applied this workflow to conduct a large-scale RVAS on the UKBB 470k WES dataset, where we identify novel biological discoveries using our approach (Results, Section 4, paragraphs 3-4).

These analyses demonstrate the advantage of DeepRVAT's modular design, which provides a deep-learning based impairment score model compatible with established robust association testing frameworks such as REGENIE.

R1.2 The authors only use the total number of significant associations to demonstrate the advantage of the proposed method. These claims would not be necessarily helpful. One would want to see how many putatively new genes or masks of rare variants can be detected using each method. To identify putatively new rare variant associations, one possible way is to perform conditional analysis of each significant association by adjusting for known variants nearby.

We have addressed this comment in two ways.

First, we have now used results from existing GWAS to control for common variants in the vicinity of the gene under consideration. We have taken a conservative approach, including independent common variants (MAF > 1%, LD clumping; Methods) within 500 kb around the gene boundaries with suggestive evidence for significance ($P < 10^{-7}$) as covariates in the association testing step. Consistent with previous reports of such conditioning^{2,3}, DeepRVAT association results with and without conditioning are markedly consistent (Results, Section 2, paragraph 5; Supp. Fig. 2.7; with conditioning resulting in a 2.6% reduction in the total number of discoveries). Notably, DeepRVAT with conditioning still yields a

larger number of discoveries than alternative methods run without such a control (Supp. Fig. 2.7b).

Second, we present results from applying DeepRVAT to the full 470k WES release of the UKBB (c.f. response to comment R1.1), both with and without conditioning on common variants (Table 1). We report previously unknown associations, for several of which we do identify complementary evidence suggesting that these are genuine novel discoveries enabled by our approach. These findings are discussed in Results (Section 4, paragraphs 3 & 4).

R1.3 This reviewer did not observe any inflation or deflation of STAAR for quantitative from the Q-Q plot in Fig.3d and Supp. Fig. 3.3a. It is not meaningful to use the genomic inflation factor λ to evaluate the calibration of STAAR. STAAR uses the ACAT method to integrate the p-values in the weighting scheme to incorporate multiple functional annotations, and the ACAT method gives accurate p-values for the tail probability⁶.

Thank you for this comment. We agree that the inflation factor λ does not reflect the relevant aspect of calibration (tail probabilities) well. We now show QQ plots throughout (Fig. 2b), which indeed indicates that all methods are calibrated on quantitative traits.

The point does hold, however, for binary traits. The challenges to obtain calibrated association results on (imbalanced) binary traits are now examined in more detail in Fig. 4b-c and Supp. Fig. 4.1, where we observed considerable benefits of combining DeepRVAT with REGENIE (c.f. response to comment R1.1).

R1.4 Following (3), as Fig. 3a showed that STAAR detected slightly more significant associations than DeepRVAT for 21 training traits. This reviewer recommends that the author perform more analysis to provide additional evidence or only claim that the proposed method has comparable performance as STAAR for quantitative traits.

We agree that the number of discoveries was comparable between DeepRVAT and STAAR in Fig. 3a of the original manuscript. In response to reviewer comments concerning the training procedure and its dependency on the selection of seed genes, relatedness, and multiple testing correction, we have now made several refinements to DeepRVAT and the benchmarking procedures. As a side effect of these changes, the performance differences with respect to other methods are now somewhat clearer. For details on the changes, please refer to the response preamble.

Having said this, we agree that minor differences in the number of gene discoveries for quantitative traits is not the major added value of our method. The revised manuscript now places more weight on the benefits of combining

DeepRVAT with REGENIE and on the biological interpretation of the new gene discoveries for binary traits obtained on the full UKBB WES release (new main text Table 1; Results, section 4, paragraph 4).

R1.5 The results of significant gene-trait associations (Supp. Table 6) are misleading. Many of the “new DeepRVAT discovery” findings are known and previously reported. Specifically, nearly all claimed “new DeepRVAT discovery” findings of lipid traits are known lipid associated genes. For example, the p-value of the association between RVs in NPC1L1 and DeepRVAT is 5.59E-07. However, the STAAR p-value of disruptive missense RVs or missense RVs or plof and disruptive missense RVs in NPC1L1 are all significantly associated with LDL using UK Biobank 200k WES data, with p-values 1.78E-07, 5.20E-09, and 3.97E-09 (see Supplementary data 14 in Selvaraj et al⁷). I am a bit confused why the authors claimed that other methods could not detect these genes.

We apologize for the lack of clarity in the presentation of these results. For reference, we had used the terminology “new DeepRVAT discovery” to differentiate between associations that were not included in the set of seed genes to train the model. The need to differentiate between seed genes and non-seed genes during testing no longer applies thanks to the substantially revised training and testing procedure of the DeepRVAT model (cf. preamble). These changes allow a more direct like-with-like comparison between the results from DeepRVAT and alternative methods.

In addition to revising this specific table and the presentation of results in the benchmarking part of the paper, we have now applied DeepRVAT to the full WES release of UKBB. On this larger dataset, we have analyzed individual discoveries at a greater level of detail (cf. Table 1), which points to discoveries that, to the best of our knowledge, have not previously been reported. We discuss several novel gene trait associations in the main text (Section 4, paragraph 4).

Other comments

R1.6 Following (5), The results of significant gene-trait associations (Supp. Table 6) are not clear. These results could not be claimed as new discoveries, as many of the results are not well calibrated for the necessary covariates. For example, in addition to age, sex and ancestry PCs, lipid traits need to account for the statin usage^{7,8}. This comment is also related to the main concern 2a. The total number of significant findings in the manuscript is not biologically meaningful.

We have revised the association tests for lipid traits to correct for statin usage as described in [7]. We agree that the total number of discoveries per se is not biologically meaningful as this number can be inflated for non-calibrated models. As mentioned above, we now provide a more stringent cross-validation training scheme, and combine DeepRVAT with REGENIE for the analysis of binary traits.

We show that our method has good calibration. Furthermore, and as mentioned in response to the preceding point, we now provide more in-depth discussion of biologically novel findings, focusing on binary traits in the full UKBB WES release. This shows biologically plausible and interesting associations.

Reference

1. Taliun, D. et al. Sequencing of 53,831 diverse genomes from the NHLBI TOPMed Program. *Nature* 590, 290-299 (2021).
2. Investigators, A.o.U.R.P. The “All of Us” research program. *New England Journal of Medicine* 381, 668-676 (2019).
3. Li, X. et al. Dynamic incorporation of multiple in silico functional annotations empowers rare variant association analysis of large whole-genome sequencing studies at scale. *Nature genetics* 52, 969-983 (2020).
4. Mbatchou, J. et al. Computationally efficient whole-genome regression for quantitative and binary traits. *Nature Genetics* 53, 1097-1103 (2021).
5. Backman, J.D. et al. Exome sequencing and analysis of 454,787 UK Biobank participants. *Nature* 599, 628-634 (2021).
6. Liu, Y. et al. ACAT: a fast and powerful p value combination method for rare-variant analysis in sequencing studies. *The American Journal of Human Genetics* 104, 410-421 (2019).
7. Selvaraj, M.S. et al. Whole genome sequence analysis of blood lipid levels in >66,000 individuals. *Nature Communications* 13, 5995 (2022).
8. Hindy, G. et al. Rare coding variants in 35 genes associate with circulating lipid levels—A multi-ancestry analysis of 170,000 exomes. *The American Journal of Human Genetics* 109, 81-96 (2022).

Reviewer #2:

Remarks to the Author:

Clarke, Holtkamp, et al. propose DeepRVAT, a data-driven deep learning neural network for rare variant analyses built from a variety of variant annotations. DeepRVAT essentially computes a gene impairment score by integrating a phenotype module and a dynamic set of variant annotations, which accounts for the nonlinear effects of rare variants, imbalanced case-control ratio of binary traits and applies to multiple phenotypes. With the gene impairment score, DeepRVAT not only improves gene discoveries through their adaptive rare variant association tests but also aids in phenotype prediction that incorporates rare variant effects. DeepRVAT appears to be more computationally efficient enough to be applied to biobank level data, which would facilitate the downstream analysis of rare variants from a broader perspective and at a larger scale.

The manuscript is well-organized and clearly written. The data and results provided in the text adequately demonstrate the accuracy and efficiency of DeepRVAT compared to the previous RVAS methods. This new method would serve as a valuable complement to the

traditional burden tests and approaches for PRS-based phenotype prediction, in that it effectively incorporates more information from variant annotations and can be highly generalizable in terms of both variant and phenotype information, which will help advance the understanding of rare variants and complex disease. Here are some comments that I hope would improve the manuscript:

Major comments:

R2.1. I applaud the authors for building a simulation framework that attempts to balance many parameters in simulating true signals among noise. However, when digging into Supp. Fig. 2.1, it seems that LoF-like variants (e.g. frameshifts) seem quite a bit underspecified and have fewer proportion causal than missense. Are the effect sizes modeled more strongly among pLOF variants as one would expect? Put another way, when running the burden tests on these simulated variants, are results comparable to those in UK Biobank which show increased signal among pLOF variants above missense? This is somewhat observed in Fig. 2a where the significance of pLOF is lower than missense and in 2b where the missense burden is higher powered than the pLOF, which is inconsistent with the Backman and Genebass analyses where most signals stem from pLOF. That they get relatively higher importance values in the real data (Supp. Fig. 3.5) is encouraging, but the simulation framework seems a bit misspecified in this regard.

It is correct that the simulations were not designed for having realistic effect sizes per variant category. Instead, the primary aim of the simulation was to validate the model and specific properties, most importantly the ability of the model to learn ground truth effects and meaningful annotation filters from data, rather than requiring predefined (and matching) cutoffs. This feature is illustrated by varying the simulation and analysis parameters of minor allele frequency, and holds irrespective of the extent of biological realism of the simulation parameters. We have included a note in the description of the simulation (Methods, section 7.2) to clarify its purpose.

As pointed out by this reviewer, the key benchmark has to be in the context of applications to real data. Given the increased emphasis of the revised manuscript on biological discovery and the application to the 470k UK Biobank WES dataset (new Results section 4 and Table 1), we have moved the results from the simulation study to a supplementary note, as well as reduced the emphasis on their analysis in favor of more detailed analysis on real data. These results are now displayed in Supp. Figs. 1.3-1.6.

R2.2. Similarly, it may be advantageous to use synonymous variants as a sort of negative class. Of course, some synonymous variants may be associated due to LD, but this might give some level of baseline expectation.

We agree that this is an interesting control to consider. We have included an analysis of the RVAT association step when applying the pre-trained gene impairment module to compute impairment scores on synonymous variants only (Supp. Fig. 2.3), which resulted in only 4 discoveries (FWER < 5%) vs. 272 when applying DeepRVAT to all variants (21 traits, 407,148 tests, total).

R2.3. The authors mentioned in the Discussion section that it is challenging to choose the most appropriate set of annotations. Would it be possible for the authors to provide some general recommendations for phenotype and variant annotation selection, as well as parameter configurations (number of repeats, number of seed genes, etc.) to users who would want to customize their own model?

We agree that this is necessary practical information for users and have provided the recommendations as part of Section 8 in the Methods and as part of the improved DeepRVAT package documentation. In addition, we have provided analyses, in Supp. Figs. 2.4, 2.5 and 2.8 on the effects of training data selection and model architecture on DeepRVAT results.

R2.4. How are the association results for the seed genes and target phenotypes used in training the gene impairment scores obtained using DeepRVAT, if they cannot be included in the association testing module?

We have revised the training and association testing to use a cross-validation scheme (cf. preamble). Using this scheme, it is now possible to use association testing results from DeepRVAT also for the seed genes and target phenotypes, since model training and gene impairment score computation are now carried out on distinct sets of samples. In this scheme, we take steps to also prevent potential information leakage due to related samples.

We have empirically confirmed that the scheme works as expected, by training DeepRVAT on seed genes with no expected association to the target phenotypes and analyzing association testing results on these genes (Supp. Fig. 2.5).

R2.5. Given that the seed genes are selected from those with associations with phenotypes of interest, are the seed genes supposed to be associated with all selected phenotypes or any phenotype from the target set? Also, will there be a default set of seed genes provided for users?

Thank you for bringing this unclear exposition to our attention. We have clarified (Results, section 1, paragraph 2; Methods, section 1.4) that seed genes should be associated with a single phenotype. That is, a prediction for a given phenotype is made based on variants in seed genes *for that phenotype*.

Regarding a default set of seed genes, it is possible to leverage independent data to select seed genes. In practice, however, we recommend choosing seed genes from those associated to a phenotype based on conventional methods applied to the training dataset. The reason is that the profile of rare variants differs across datasets, and to be a useful seed gene, the association must be present based on the rare variant profile available in the specific training dataset itself. A discussion of this point has been added to Methods (section 8) and in the DeepRVAT package documentation.

We note here that the model is empirically robust to the specific choice of seed genes (Supp. Figs. 2.4, 2.5).

R2.6. The description of the model is a bit confusing at times. Figure 1 suggests that only annotations are used in the gene impairment model, but later “We used 21 quantitative traits [...] to train DeepRVAT”. Can you clarify this, and if indeed phenotypes are used, elaborate more on how this model can be generalized to any other phenotypes with no restrictions, more specifically, what makes the gene impairment score both gene- and trait-agnostic?

We have expanded the caption of Fig. 1 to better describe the input data and training procedure, as well as revised Section 1.4 of the methods. The core assumption to achieve transferability is that the parameters of the gene impairment module are shared across all genes and traits.

Additionally, we have identified the results which we believe support the statement that the gene impairment score is gene- and trait-agnostic, namely the application, in association testing, to genes and traits on which the model has not been trained (Results, section 2, paragraph 5).

R2.7. How will DeepRVAT generalize to cohorts with multiple ancestry groups?

We have integrated DeepRVAT into REGENIE, and use its approach of including covariates for polygenic effects to control for relatedness and population structure. These results are presented in Fig. 4. Please see the preamble and the response to comment R1.1 for more details.

R2.8. The authors note that “the overall contribution of rare variants was modest” which is very much in line with expectations. Have they considered this separately for each phenotype? Specifically, there are new methods that estimate heritability contribution from rare variants (Weiner, Nadig, et al, 2023) - are the contributions from the rare variants in this model correlated with burden heritability results?

We have analyzed the correlation between the added R^2 for DeepRVAT (Supp. Table 8) and burden heritability and find a substantial ($r = 0.7$) correlation between

the two (Supp. Fig. 3.1; Results, section 3, paragraph 2). This shows, reassuringly, that the performance of DeepRVAT scales with the global measure of burden heritability.

R2.9. Supp. Fig. 3.1: I'm a bit surprised that MAF is not correlated with other features. Is this an artifact of Pearson correlation being not ideal for a linearly compressed metric like MAF? Would Spearman correlation (or $\log_{10}(\text{AF})$) be better here?

We have modified the correlation heatmap (now Supp. Fig. 2.2) to use Spearman correlation, and have included $\log_{10}(\text{MAF})$ as an additional annotation in the heatmap. We still don't see a very strong correlation with other features, which we also find surprising, but it is what the data shows.

R2.10. Similarly, there seems to be a group of highly correlated features included in model training. Can the authors comment on how such inclusion will impact the performance of the model?

We have run an analysis where we drop certain highly correlated features during model training (Supp. Fig. 2.4; Results, section 2, paragraph 4). The model is fairly robust to these dropouts, but notably performs best when all features are included. Therefore, we recommend to include a broad collection of available annotations (c.f. Methods 8 and in the DeepRVAT package documentation).

R2.11. In Supp. Fig. 3.7 and Methods 3.2, when repeating for more than one time, what are the parameters passing into the next repeat? If there are not any elements inherited, how are the uncorrected p-values from each repeat being processed and aggregated?

We have addressed this point in the context of the new training procedure.

Rather than aggregating at the level of p-values, we now average gene impairment scores across repeats before association testing, so that a single p-value is computed for each gene. Downstream, this is simpler than in the previous procedure. To aggregate p-values across genes, we apply Bonferroni correction.

There is no information passed from one repeat of model training to the next - each run is independently stochastically initialized.

R2.12. Why is DeepRVAT specific to rare variants? Are there any assumptions that prevent applying it to common variants?

Since submitting the paper, we have begun working on extending DeepRVAT to both common and rare variants. There are, to the best of our knowledge, no *a priori* reasons not to apply DeepRVAT to common variants as well. However,

based on our initial results, we believe that the best results can be achieved by treating common variants differently from rare variants since, as in GWAS vs. RVAS, their effects may be best captured not by their annotations but by their simple statistical association with phenotype, which requires notably fewer assumptions.

We have added a brief note on future directions towards joint modeling of common and rare variants to the Discussion (paragraph 3).

Minor comments:

1. The authors claimed that DeepRVAT presents well-calibrated results for imbalanced binary traits. It would be helpful to report the number of cases and controls for the binary traits analyzed in this study and compare the performance of different models across different case/control ratios to validate the statement.

Case/control counts have now been added to Supp. Table 3, and the prevalence is shown on the x-axis of the new Fig. 4c.

2. The authors should clarify the default proportion of causal variants for simulation in Figure 2 in the main text. Similarly, in Figure 2a, it is hard to see the causal variants among the non-causal ones. Comparing the mean chi-squared (or similar) for causal vs non-causal may make this point more strongly.

We have clarified the default proportion of causal variants in the figure legend (now Supp. Fig. 1.5). In the QQ plots, we show P-values for all genes, colored by causal and non-causal ones. In addition to computing lambda GC for all genes, we've also included lambda GC values specifically for causal and non-causal genes in the figure.

3. It took me a while to understand what the rows in Figure 2b represented - perhaps a simplified label (high, medium, no influence of ultra-rare variants, or something like that) would help orient.

From bottom to top, increasing relevance gets attributed to variants with larger MAF. For instance, in the bottom row, no variant from the largest MAF bin (0.1%-1%) is causal, while in the top row, they explain ~20% of the cumulative variant effect. To enhance clarity, we've included an arrow indicating the increasing relevance of variants with larger MAF from bottom to top.

4. The authors should add or cite results related to this statement "... including in settings for which additional non-causal variants were incorporated in the MAF cutoff..." in the "Model validation using simulated data" section.

We have clarified the results to which this statement refers. In the bottom-right panel of Supp. Figure 1.5, all methods analyze variants with up to a 1% MAF for association testing. However, in this panel, none of the variants within the largest MAF bin (0.1%-1%) are causal, suggesting they contribute as 'noise' to the models. Conventional methods show notably low power in this scenario (Supp. Figure 1.5, bottom-right). As the MAF filter aligns better with simulated causal variants, the performance of all methods improves (bottom row, right to left). However, the disparity in performance between a well-aligned MAF filter (bottom-left, association testing MAF < 0.01%) and a poorly aligned one (bottom-right) is much more pronounced for conventional methods than for DeepRVAT.

5. Were the traits used for training chosen in some non-random way? They seem to have on average more associations (Fig. 3a) than those for evaluation (3e). I would understand if there's some light overfitting happening, but the Burden/SKAT combined also shows lower numbers, which doesn't have nearly as much influence of the traits used. Additionally, can the authors report replication results for the evaluation traits used in Fig 3d-3f (with a similar display as in Fig 3b)?

Regarding the choice of training traits, we have clarified this in Section 4.5 of the Methods. Training traits were chosen based on having sufficient seed genes for effective model training, which is why they have more associations on average.

We now show the replication results for evaluation traits in Fig. 2g, which follow a similar pattern as for the training traits.

We have not shown replication results related to binary traits, since the phenotype definitions we use from Jurgens et al. differ substantially from those used in Backman et al. and Genebass^{2,4,5}.

6. The authors should label the panels of Figure 4b and 4c.

Thank you for pointing out this omission. We have now included labels (now Fig. 3b, c).

7. What are the dotted lines in Figure 4c?

The meaning of the dotted lines has been clarified in the figure caption (now Fig 3c).

8. Supp. Table S2, instead of Supp. Table S4 should be cited in the caption of Supp. Fig. 2.1.

Thank you. This has been corrected.

9. Supp. Fig. 2.1 is missing captions for panels a and b, specifically for panel b, a short description of the metric in each panel might be helpful in explaining the different trends between causal and non-causal variants.

We have added the missing captions and addressed the specific point about the metric.

10. Supp. Fig. 2.1e is missing a color legend for the dark blue boxes.

We have corrected the legend.

11. In Supp. Fig 3.6, the authors should label the panels and avoid using “rows” and “columns” in the caption if they do not refer to the rows and columns of the figure. Is each panel specific to a VEP annotation? These should be spelled out if so, and the caption clarified.

This supplemental figure has been removed as, given the modified training procedure, it is no longer relevant.

12. Can the authors provide an explicit list of the required input files/information for each module in the Github repo of DeepRVAT?

We have improved the documentation of the package and provided the requested lists.

Reviewer #3 (Remarks to the Author)

The authors presented a novel method for rare variant association test, DeepRVAT, that is leveraging set neural networks to integrate genotype with variant annotations for optimal power for association testing. DeepRVAT also comes with a rare variant-based prediction functionality as an added feature. In the simulation study, DeepRVAT showed faster computation, good calibration of test statistics, and better power. Overall, this manuscript presented a novel application of deep neural networks to genetic association testing and showed good performance of the method for association testing and phenotype prediction. I have only a few comments on the manuscript.

My biggest concern about the method is that the model is trained based on selected sets of seed genes for the phenotypes included in the model training. This approach raises several questions:

R3.1. There is no evaluation of how the selection of seed genes affects the performance of the trained model, especially with real data analysis. It is hard to imagine that different sets of seed genes will yield models with the same performance in association testing and genetic prediction. Yet there's no mention of such concern or evaluation on this particular point (or did I overlook?) The authors also did not present any information on the seed genes in the application on UKB data. Therefore, there's no way to evaluate whether the choice of the seed genes makes sense in those analyses based on prior knowledge of the phenotypes analyzed with the UKB data. A simple test may be down-sampling the seed genes used in current UKB analysis into equal sized random sets of seed genes and see if such permutation affects the model performance.

Please see the response to the next comment.

R3.2. Extending from the previous comment, the choice of phenotypes included in model training may also affect the trained model performance. However, there's also no mention of this potential issue either.

We have carried out the suggested downsampling analyses from this and the previous comment (Supp. Fig. 2.4; Results, section 2, paragraph 4). Overall, we find that DeepRVAT's results are relatively robust to changes in the training data, with a somewhat larger effect seen when removing traits vs. removing seed genes.

As an additional test of robustness to choices in training data, we have assessed the robustness of DeepRVAT to *adding* seed genes without an expected association to any of the training phenotypes (Supp. Fig. 2.5). This analysis shows that the CV training procedure (see preamble) protects against spurious associations due to overfitting.

R3.3. As the authors did in the study, the selected seed genes can be based on results of alternative rare variant association testing methods if prior knowledge is lacking or maybe not desired. However, this strategy raises several questions on the overall benefit of using DeepRVAT:

a. What if there's no significant results from the other RVAT methods? The users are essentially left to make an (arbitrary) choice on seed genes based on prior knowledge.

Thank you for bringing up this point. We have added detail on the intended usage patterns of DeepRVAT in the main text, but particularly also in Section 8 of Methods and in improved package documentation.

For most scenarios, we recommend that users apply (1) precomputed DeepRVAT scores for UK Biobank, which we will make available via UKBB, or (2) for other

datasets, pretrained models, which are provided as part of the DeepRVAT package.

In the scenario you mentioned, we recommend usage (2) with pretrained models (see also the response to comment R2.5). The generalizability of the pretrained models to traits and individuals not seen during training has been demonstrated via the application to quantitative (Fig. 2f-g; Results, section 2, paragraph 5) and binary traits (Fig. 4b-d and Table 1; Results, section 4) not seen during training, as well as the applications to individuals held out during training (Fig. 4a and Table 1. Results, section 4).

b. DeepRVAT does not take into account for leveraging the knowledge of known associated genes in model training. The multiple testing burden for DeepRVAT should be, in my opinion, considering all association tests performed with the other RVAT methods used to identify the seed genes plus the association tests performed with DeepRVAT. Implicitly, that is how many tests DeepRVAT performed to reach its conclusion on association findings. Even if the seed genes are selected based on prior knowledge (without additional RVAT analysis), it still implicitly leverages prior analysis results (just in a way that is hard to ascertain the multiple testing burden).

Thank you for bringing up this subtle but important point.

In response to this and other reviewer comments, we have revised the training procedure of DeepRvat and taken an even stricter and arguably statistically cleaner approach to the seed gene selection step. The new CV training procedure (cf. preamble) effectively addresses the risk of leakage of information between seed gene selection and the final association tests. In particular, this procedure prevents double-dipping, as gene impairment scores for any given individual in the cohort are computed using a model that was not trained on that individual (or any related individuals).

Most importantly, we have carried out extensive empirical analyses that give us confidence that DeepRVAT is robustly calibrated (Fig. 2b, c, g, Fig. 4b-c, Supp. Figs. 2.3 and 4.1), and in particular, we can rule out that including a gene as a seed gene during model training leads to inflated test statistics (Supp. Figs. 2.4 and 2.5, see also responses to Comments R3.1 and R3.2 above).

c. Also, the computation burden should actually consider all tests performed, including the other RVAT methods, if that is a necessary step for DeepRVAT to work.

This had already been accounted for in the figure, however, thank you for pointing out that this had not been mentioned anywhere. We have revised the caption to make this explicit.

d. A minor point is that, as the users are left with results from alternative RVAT method(s) and DeepRVAT, it could be confusing that these results from different methods by default cannot replicate each other. First, DeepRVAT cannot validate the observed significant associations of the seed genes, which showed significant association from other methods, since the DeepRVAT model is trained on them. On this point, the users are also left with an incomplete picture of the phenotype-gene associations from DeepRVAT, i.e. the users would not be able to compare the associations between the seed genes and other gene findings from DeepRVAT. On the other hand, all the new findings in DeepRVAT are by default not significant from the other methods. Technical replication across methods is just not possible in this case. (However, replication using different samples is possible like what the authors did in the paper.)

With the new CV training and association testing scheme, all genes are tested using DeepRVAT, with conventional methods only used to determine training (seed) genes for DeepRVAT where required. Thus, technical replication between DeepRVAT and methods used during seed gene discovery is now possible. Of course, one might worry that DeepRVAT associations on seed genes might be inflated due to overfitting, thus leading to false positive technical replication. However, as noted, we have empirically shown in Supp. Fig. 2.5 that this tends not to occur.

Additionally, when using precomputed burdens or pretrained models on traits and/or samples not seen during DeepRVAT training, as we recommend in most cases, this concern does not apply. We now place substantially more emphasis on these use cases. See revised main text figures Fig. 2f, g & Fig. 4. Notably, the replication rates are very similar for application of the model to training traits (Fig. 2c) versus novel traits not used during training (Fig. 2g).

It may be the case that the methodological advances of DeepRVAT in general overcomes the potential impact of training data quality on the trained model's performance, i.e. no matter how the training data perturb, the DeepRVAT will perform similarly well compared to other methods. If that is the case, it is actually a great advantage of DeepRVAT. However, I would suggest the authors show it in real data analysis examples.

Thank you for bringing up these important points. We hope that the analyses laid out above (particularly Supp. Figs. 2.4 and 2.5) have addressed this. Robustness of the gene impairment module to perturbations of the training data is exactly what we see.

Minor comments:

1. The effect size estimates from DeepRVAT is on the scale of gene impairment score, which is of unknown interpretation. The users may eventually need to run burden tests for an interpretable effect size if interested. May include this in discussion.

We have ensured that DeepRVAT scores are on an interpretable scale between 0 and 1. However, we do agree that even though pLOF variants do tend to be scored near the high end of the distribution (Fig. 2d), the effect size is not as interpretable as the "number of missense variants" or "number of pLOF variants" that might be used as a conventional burden score.

While we agree that this point requires mentioning, we thought it was a little technical to try to fit into the limited space of the Discussion. However, we have added it to Section 8 of the Methods and in the updated package documentation.

2. In Supplementary Methods 3.3 "Controlling for overfitting", section "Gene-gene correlations", the authors said that "Biologically, we have no reason to expect that the existence of, say, a pLOF variant in one gene should correlate with the existence of such a variant in another gene in the same individual." However, for genes nearby or with overlapping exons, coding variants could still be correlated across genes (theoretically at least). I would guess the low correlation shown in Supp. Fig. 3.6 is due to low LD between rare variants. May include this in the section.

Thank you for pointing out this important subtlety. However, with the introduction of the CV scheme for training and association testing, this section of the Methods has become obsolete and was removed.

Decision Letter, first revision:

13th Jun 2024

Dear Oliver,

Thank you for submitting your revised manuscript "Integration of variant annotations using deep set networks boosts rare variant association genetics" (NG-TR62980R2). It has now been seen by the original referees and their comments are below. The reviewers find that the paper has improved in revision, and therefore we'll be happy in principle to publish it in Nature Genetics, pending minor revisions to satisfy the referees' final requests and to comply with our editorial and formatting guidelines.

As the current version of your manuscript is in a PDF format, please email us a copy of the file in an editable format (Microsoft Word or LaTeX)-- we can not proceed with PDFs at this stage.

Sincerely,

Michael Fletcher, PhD
Senior Editor, Nature Genetics
ORCID: 0000-0003-1589-7087

Reviewer #1 (Remarks to the Author):

The authors have fully addressed the reviewer's comments. This reviewer particularly appreciates the efforts made to expand the application of DeepRVAT for biological discovery and the extended analysis and assessment of DeepRVAT on (imbalanced) binary traits. My remaining concerns are minor:

(1) The statement "either by conducting a meta-analysis over different test types and annotations" in the introduction section is not accurate. The STAAR method integrates various test types and multiple functional annotations through an omnibus test, not a meta-analysis. The approach that incorporates a meta-analysis within the STAAR framework is called MetaSTAAR1. To avoid confusion, it is recommended to use the following: "Recently proposed RVAT methods based on variance component tests convincingly demonstrated the added value of incorporating a broad spectrum of annotations, either by conducting an omnibus test over different test types and annotations^{1,2}, or using specialized kernels tailored to different annotation types³ (Methods; Supp. Table 1)."

Reference

1. Li, X. et al. Powerful, scalable and resource-efficient meta-analysis of rare variant associations in large whole genome sequencing studies. *Nature Genetics* 55, 154-164 (2023).
2. Li, X. et al. Dynamic incorporation of multiple in silico functional annotations empowers rare variant association analysis of large whole-genome sequencing studies at scale. *Nature Genetics* 52, 969-983 (2020).
3. Monti, R. et al. Identifying interpretable gene-biomarker associations with functionally informed kernel-based tests in 190,000 exomes. *Nature Communications* 13, 5332 (2022).

Reviewer #2 (Remarks to the Author):

The authors have satisfied all of my concerns.

Reviewer #3 (Remarks to the Author):

I really appreciate the authors' revision to the DeepRVAT method. In particular, the authors proposed a cross validation scheme for training the gene impairment module with a revised seed genes selection approach. The additional real data analysis and biological discovery comes with the application of DeepRVAT also added value to the manuscript, makes it goes beyond method development and showed the potential of the DeppRVAT. I do not have further comments on the manuscript.

Final Decision Letter:

20th Aug 2024

Dear Oliver,

I am delighted to say that your manuscript "Integration of variant annotations using deep set networks boosts rare variant association testing" has been accepted for publication in an upcoming issue of Nature Genetics.

Your paper will be published online after we receive your corrections and will appear in print in the next available issue. You can find out your date of online publication by contacting the Nature Press Office (press@nature.com) after sending your e-proof corrections.

Acceptance is conditional on the data in the manuscript not being published elsewhere, or

announced in the print or electronic media, until the embargo/publication date. These restrictions are not intended to deter you from presenting your data at academic meetings and conferences, but any enquiries from the media about papers not yet scheduled for publication should be referred to us.

Please note that *Nature Genetics* is a Transformative Journal (TJ). Authors may publish their research with us through the traditional subscription access route or make their paper immediately open access through payment of an article-processing charge (APC). Authors will not be required to make a final decision about access to their article until it has been accepted. Find out more about Transformative Journals

Authors may need to take specific actions to achieve compliance with funder and institutional open access mandates. If your research is supported by a funder that requires immediate open access (e.g. according to Plan S principles) then you should select the gold OA route, and we will direct you to the compliant route where possible. For authors selecting the subscription publication route, the journal's standard licensing terms will need to be accepted, including <https://www.nature.com/nature-portfolio/editorial-policies/self-archiving-and-license-to-publish>. Those licensing terms will supersede any other terms that the author or any third party may assert apply to any version of the manuscript.

If you have not already done so, we strongly recommend that you upload the step-by-step protocols used in this manuscript to protocols.io. protocols.io is an open online resource that allows

researchers to share their detailed experimental know-how. All uploaded protocols are made freely available and are assigned DOIs for ease of citation. Protocols can be linked to any publications in which they are used and will be linked to from your article. You can also establish a dedicated workspace to collect all your lab Protocols. By uploading your Protocols to protocols.io, you are enabling researchers to more readily reproduce or adapt the methodology you use, as well as increasing the visibility of your protocols and papers. Upload your Protocols at <https://protocols.io>. Further information can be found at <https://www.protocols.io/help/publish-articles>.

Sincerely,

Michael Fletcher, PhD
Senior Editor, Nature Genetics
ORCID: 0000-0003-1589-7087